# The transcription factor Rreb1 regulates epithelial architecture, invasiveness, and vasculogenesis in early mouse embryos

Sophie M Morgani[1,2]*, Jie Su[3], Jennifer Nichols[2], Joan Massagué[3], Anna-Katerina Hadjantonakis[1]*

[1]Developmental Biology Program, Sloan Kettering Institute, Memorial Sloan Kettering Cancer Center, New York, United States; [2]Wellcome Trust-Medical Research Council Centre for Stem Cell Research, University of Cambridge, Jeffrey Cheah Biomedical Centre Cambridge Biomedical Campus, Cambridge, United Kingdom; [3]Cancer Biology and Genetics Program, Sloan Kettering Institute, Memorial Sloan Kettering Cancer Center, New York, United States

**Abstract** Ras-responsive element-binding protein 1 (Rreb1) is a zinc-finger transcription factor acting downstream of RAS signaling. *Rreb1* has been implicated in cancer and Noonan-like RASopathies. However, little is known about its role in mammalian non-disease states. Here, we show that Rreb1 is essential for mouse embryonic development. Loss of *Rreb1* led to a reduction in the expression of vasculogenic factors, cardiovascular defects, and embryonic lethality. During gastrulation, the absence of *Rreb1* also resulted in the upregulation of cytoskeleton-associated genes, a change in the organization of F-ACTIN and adherens junctions within the pluripotent epiblast, and perturbed epithelial architecture. Moreover, *Rreb1* mutant cells ectopically exited the epiblast epithelium through the underlying basement membrane, paralleling cell behaviors observed during metastasis. Thus, disentangling the function of Rreb1 in development should shed light on its role in cancer and other diseases involving loss of epithelial integrity.

*For correspondence:
sophie.morgani@nyulangone.org (SMM);
hadj@mskcc.org (A-KH)

Competing interests: The authors declare that no competing interests exist.

## Introduction

Ras-responsive element-binding protein 1 (RREB1) is a zinc-finger transcription factor that acts downstream of RAS (*Thiagalingam et al., 1996*). It is evolutionarily conserved (*Ming et al., 2013*), widely expressed (*Fujimoto-Nishiyama et al., 1997*), can function both as a transcriptional repressor and activator (*Deng et al., 2020*), and interacts with several signaling pathways, including EGFR/MAPK (*Kim et al., 2020*) and JNK/MAPK (*Melani et al., 2008*; *Reed et al., 2001*), which regulate RREB1 through phosphorylation, and JAK/STAT (*Melani et al., 2008*), TGF-β/SMAD (*Su et al., 2020*), Notch, and Sonic Hedgehog (*Sun and Deng, 2007*), which cooperate with RREB1 in transcriptional regulation. These properties suggest that RREB1 plays key contextual biological roles.

In humans, Rreb1 acts as a transcriptional repressor of HLA-G, a secreted factor that mediates vascular remodeling and tumor cell immune evasion (*Flajollet et al., 2009*; *Liu et al., 2020*). Moreover, mutation or altered expression of Rreb1 has been linked to leukemia (*Yao et al., 2019*), melanoma (*Ferrara and De Vanna, 2016*), thyroid (*Thiagalingam et al., 1996*), and prostate (*Mukhopadhyay et al., 2007*) cancers, as well as pancreatic and colorectal cancer metastasis (*Cancer Genome Atlas Research Network. Electronic address and Cancer Genome Atlas Research Network, 2017*; *Hui et al., 2019*; *Kent et al., 2017*; *Li et al., 2018*). Additionally, loss of a single allele of *Rreb1* causes Noonan-like RASopathies in adult mice, including craniofacial and cardiovascular defects (*Kent et al., 2020*). Thus, unraveling the varied roles of Rreb1 is of critical importance to understand its mechanism of action in disease states.

Nevertheless, we currently know little about the function of mammalian Rreb1 in normal, non-disease states. In *Drosophila*, the homolog of Rreb1, *Hindsight* (hnt, also known as *pebbled*), is required for embryonic development (*Wieschaus et al., 1984*) where it regulates cell-cell adhesion and collective migration in various contexts, including trachea and retinal formation, border cell migration, and germ-band retraction (*Melani et al., 2008*; *Pickup et al., 2002*; *Wilk et al., 2000*). We recently reported that chimeric mouse embryos containing *Rreb1* mutant cells also exhibit early embryonic phenotypes (*Su et al., 2020*), suggesting that *Rreb1* has a role in mammalian development.

Here, to investigate this further, we generated and characterized a *Rreb1* mutant mouse line. We found that *Rreb1* is expressed within the embryo proper and extraembryonic supporting tissues and regulates a variety of processes including neural tube closure and cardiovascular development. In gastrulating mouse embryos, loss of *Rreb1* resulted in a change in the transcription of numerous factors that are typically secreted by the visceral endoderm (VE), the HLA-G homolog H2-Q2, and numerous cytoskeleton-associated genes. We observed altered organization of F-ACTIN and adherens junctions and a loss of epithelial structure within the VE and pluripotent epiblast epithelium. Furthermore, in chimeric embryos, a fraction of *Rreb1*$^{-/-}$ epiblast cells breached the underlying basement membrane and aberrantly exited the epithelium seeding ectopic cells throughout the embryo. These data demonstrated that *Rreb1* is required to maintain epithelial architecture during mammalian development and its loss promotes cell behaviors reminiscent of those in metastasis. Thus, future studies to unravel the tissue-specific targets and mechanism of action of *Rreb1* during development may also shed light on its role in disease states.

## Results

### *Rreb1* is expressed as cells exit primed pluripotency

We characterized the expression pattern of *Rreb1* during early mouse development using whole-mount preparations of embryos harboring a LacZ-tagged transcriptional reporter (*Figure 1—figure supplement 1A*, European Conditional Mouse Mutagenesis Program) (*Bradley et al., 2012*). At pre-implantation stages (embryonic day (E) 4.5), *Rreb1*$^{LacZ}$ was expressed within the inner cell mass (ICM), comprising epiblast cells that will generate the fetus and primitive endoderm (PrE) cells that will give rise to the endoderm of the yolk sac, and the trophectoderm that will form the placenta (*Figure 1—figure supplement 1B*). In the early post-implantation embryo (E5.5), before gastrulation, *Rreb1*$^{LacZ}$ was expressed within the PrE-derived visceral endoderm (VE) and trophectoderm-derived extraembryonic ectoderm (ExE), but not the epiblast (*Figure 1A*). Subsequently, during gastrulation (E6.5–7.5), expression was observed within the VE, primitive streak, a region where cells undergo an EMT and start to specify and pattern the mesoderm and endoderm germ layers, embryonic and extraembryonic mesoderm (derived from the primitive streak), and distal anterior epiblast (*Figure 1A*, *Figure 1—figure supplement 1C*). Around midgestation (E8.0–10.5), *Rreb1*$^{LacZ}$ was expressed within the yolk sac endoderm, node, notochord, primitive streak, blood, allantois, head mesenchyme, and pharyngeal arches (*Figure 1A*, *Figure 1—figure supplement 1D–G*). We noted that at E10.5 *Rreb1*$^{LacZ}$ was expressed in regions of high FGF signaling activity (*Morgani et al., 2018b*), including the limb buds, frontonasal processes, and isthmus (*Figure 1—figure supplement 1F*). The domain of *Rreb1*$^{LacZ}$ expression within the tailbud varied between embryos, suggesting that *Rreb1* transcription may be regulated by the segmentation clock (*Figure 1—figure supplement 1F*). Data were further validated by comparison to available single-cell transcriptomic (scRNA-seq) datasets of equivalent embryonic stages (*Figure 1—figure supplement 1H*; *Nowotschin et al., 2019*; *Pijuan-Sala et al., 2019*).

In vitro, the *Rreb1*$^{LacZ}$ reporter marked a subpopulation of pluripotent embryonic stem cells (ESCs) and epiblast stem cells under self-renewing conditions and became more widely expressed as cells were differentiated by removal of the cytokine LIF or addition of FGF (*Figure 1B*). Thus, *Rreb1* is initially expressed by all lineages of the pre-implantation blastocyst and is downregulated within the epiblast as it transitions from a naïve to a primed state of pluripotency. During post-implantation development, *Rreb1* continues to be expressed in extraembryonic tissues and is re-expressed in the embryonic lineages as primed pluripotency is exited and the germ layers are specified.

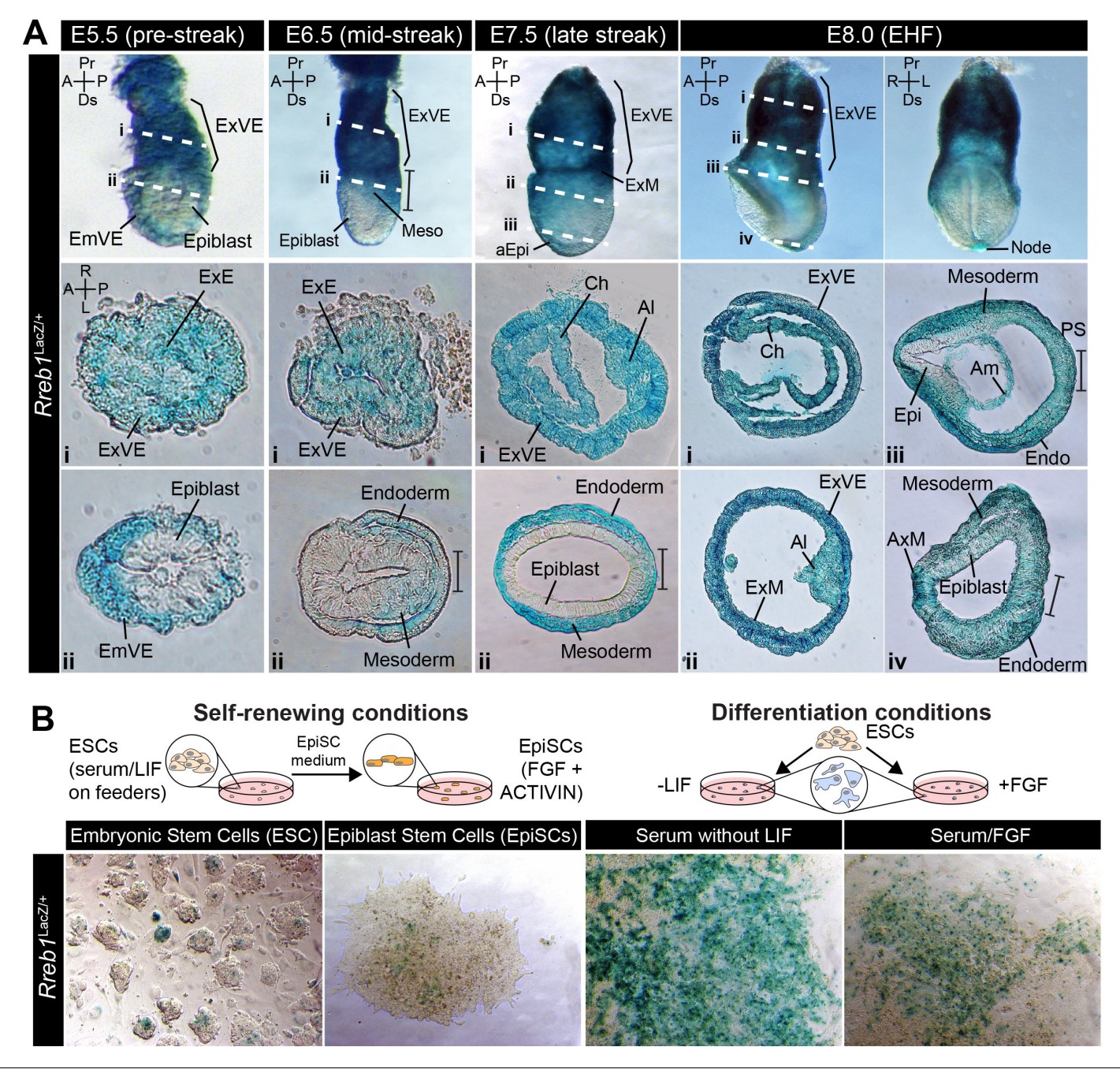

**Figure 1.** *Rreb1* is expressed within embryonic and extraembryonic tissues. (**A**) Wholemount images of *Rreb1*^LacZ/+ mouse embryos from embryonic day (E) 5.5–8.0. Dashed lines mark approximate plane of transverse sections shown in lower panels. Section iii from E7.5 is located in *Figure 1—figure supplement 1C*. Bracket demarcates the primitive streak. (**B**) *Rreb1*^LacZ reporter mouse embryonic stem cells (ESCs) (i) and epiblast stem cells (EpiSCs) (ii) under self-renewing conditions. ESCs were grown in serum/LIF on feeders. Panels (iii) and (iv) show ESCs after 7 days of differentiation in the absence of LIF or in the absence of LIF plus 12 ng/ml FGF2. A, anterior; P, posterior; Pr, proximal; Ds, distal; L, left; R, right; EHF, early headfold; ExM, extraembryonic mesoderm; ExVE, extraembryonic visceral endoderm; AVE, anterior visceral endoderm; aEpi, anterior epiblast; Meso, mesoderm; Endo, endoderm; Epi, epiblast; Am, amnion; Al, allantois; Ch, chorion; AxM, axial mesoderm.

The online version of this article includes the following figure supplement(s) for figure 1:

**Figure supplement 1.** *Rreb1* expression during mouse embryonic development.

### *Rreb1* is essential for mouse embryonic development

Previously, we generated chimeric embryos by injecting $Rreb1^{-/-}$ ESCs into wild-type host embryos. While $Rreb1^{-/-}$ cells could undergo the gastrulation EMT, migrate within the wings of mesoderm, and differentiate into germ layer derivatives, cells accumulated at the primitive streak over time suggesting that later EMT events are perturbed (*Su et al., 2020*). To further interrogate the developmental function of Rreb1, we proceeded to generate a *Rreb1* knockout mouse using CRISPR-Cas9 technology (*Figure 2A*). $Rreb1^{+/-}$ mice were viable and fertile, but heterozygous intercrosses yielded no homozygous mutant offspring. From E7.5 onwards, mutant embryos were smaller than wild-type littermates (*Figure 2B,C*, *Figure 2—figure supplement 1A,B*) and, based on morphology and somite number, were approximately 9.5 hr retarded (*Figure 2—figure supplement 1B,C*). At E9.0–9.5, $Rreb1^{-/-}$ embryos showed various defects, including microcephaly (2/7 at E9.5, *Figure 2—figure supplement 1D*), an open foregut (*Figure 2—figure supplement 1E*), and an open neural tube at

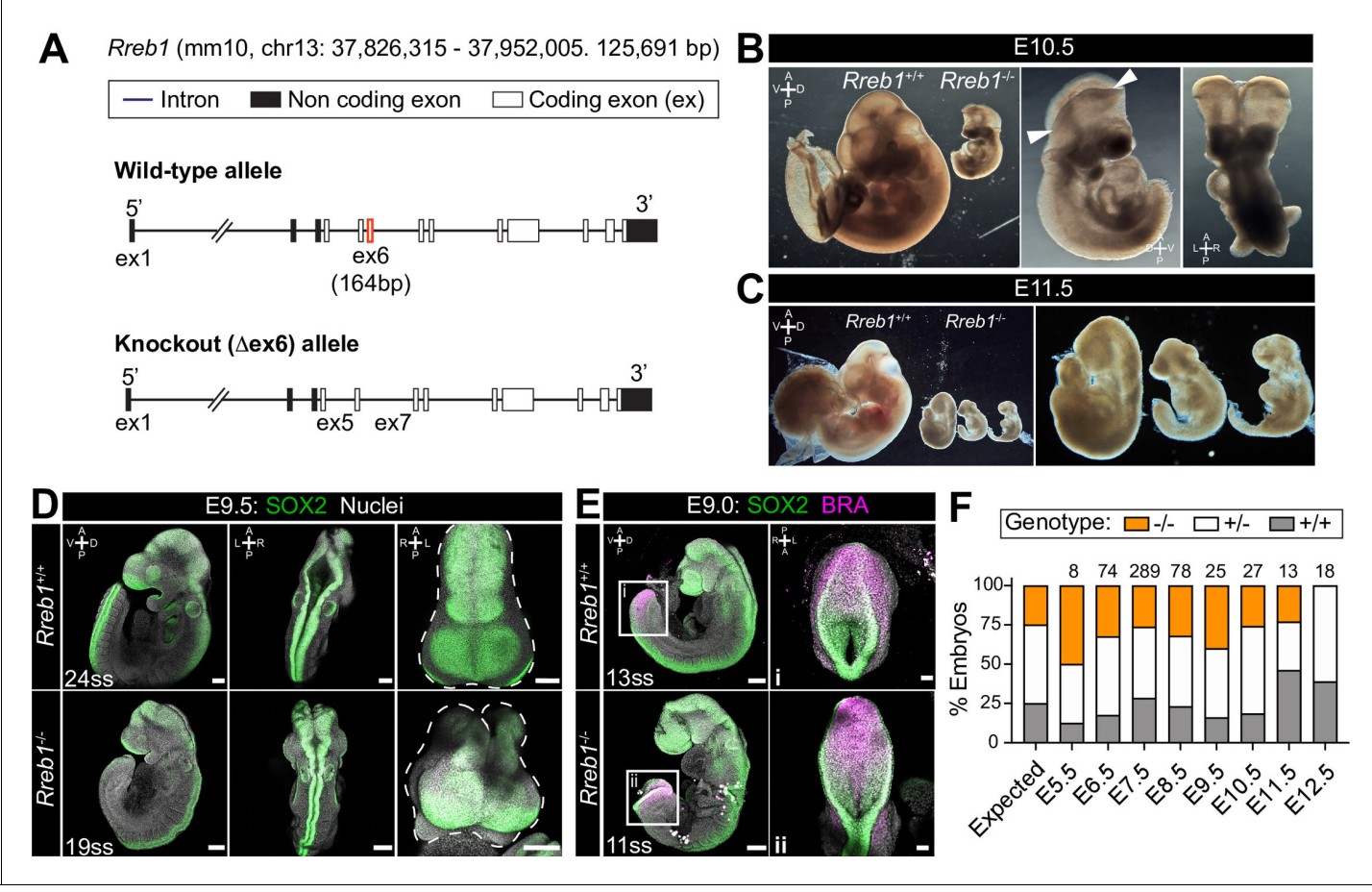

**Figure 2.** *Rreb1* is necessary for mouse embryonic development. (**A**) Schematic diagram showing the strategy used to generate the *Rreb1* mutant allele. CRISPR-Cas9 was used to delete the majority of the coding DNA sequence of Exon 6. We created a large (approximately 700 bp) and small (approximately 540 bp) deletions. Both lines exhibited comparable phenotypes, thus we combined these data. UTR, untranslated region. (**B–C**) Brightfield images of $Rreb1^{+/+}$ and $Rreb1^{-/-}$ littermates at E10.5 and E11.5 Arrowheads indicate boundary of open neural tube. Righthand panels show mutant embryos at higher magnification. (**D–E**) Confocal maximum intensity projection (MIP) of wholemount E9.0 and 9.5 mouse embryos, scale bar (sb) 200 μm. Number of somite pairs (ss) shown on images. (**D**) Right panel shows an MIP frontal view and outline (dashed line) of the head of the embryo emphasizing the neural tube closure defects in the $Rreb1^{-/-}$. (**E**) Box highlights image of posterior neuropore shown in high magnification in adjacent panel, sb 100 μm. (**F**) Bar chart summarizing the percentage of $Rreb1^{+/+}$, $Rreb1^{+/-}$ and $Rreb1^{-/-}$ embryos recovered at each developmental stage. The first bar indicates the expected Mendelian ratios of each genotype. N numbers shown above each bar. D, dorsal; V, ventral; A, anterior; P, posterior; L, left; R, right.

The online version of this article includes the following figure supplement(s) for figure 2:

**Figure supplement 1.** *Rreb1* mutant embryos exhibit defects at midgestation.

the forebrain, midbrain, and posterior neuropore level (8/10 $Rreb1^{-/-}$ at E9.5, *Figure 2D–E*, *Figure 2—figure supplement 1F*). The proportion of $Rreb1^{-/-}$ embryos with open neural tubes was reduced at E10.5 compared to E9.5 (2/8 [25%] versus 8/10 [80%] $Rreb1^{-/-}$, *Figures 2B* and *3F*), suggesting that some mutants close their neural tube upon further development.

Additionally, mutant embryos displayed aberrant notochord formation. In wild-type embryos, the axial mesoderm, marked by BRACHYURY expression in cells anterior to the gut tube, gives rise to the prechordal plate rostrally (*Figure 2—figure supplement 1G* i) and to the tube-like notochord caudally (*Figure 2—figure supplement 1G* ii-iv) (*Balmer et al., 2016*). However, in $Rreb1^{-/-}$, BRACHYURY-expressing cells did not establish a tube, instead, intercalating into the foregut (*Figure 2—figure supplement 1v*), protruding into the foregut lumen (*Figure 2—figure supplement 1G* vi), or generating multiple distinct clusters (*Figure 2—figure supplement 1G* vii). Thus, loss of $Rreb1$ results in a range of phenotypic abnormalities initiating at gastrulation and resulting in midgestation lethality.

Homozygous mutants began to be resorbed at E11.5, as marked by the disintegration of embryonic tissues (*Figure 2C*), and were not recovered at E12.5 (*Figure 2F*). Thus, $Rreb1$ is an essential factor regulating numerous processes during early mouse development.

## $Rreb1$ is required for cardiovascular development

Rreb1 is a context-dependent transcriptional repressor or activator (*Deng et al., 2020*). To define the gene expression changes associated with a developmental loss of $Rreb1$ we performed RNA-sequencing of $Rreb1^{-/-}$ embryos and compared them to wild-type ($Rreb1^{+/+}$) transcriptomes. Embryos were isolated and analyzed at E7.5 (*Figure 3A*), coinciding with the onset of morphological defects in mutant embryos (*Figure 2—figure supplement 1A,B*). We identified 65 downregulated and 200 upregulated genes in $Rreb1^{-/-}$ vs. $Rreb1^{+/+}$ embryos (fold-change >log2(2), p<0.05, *Figure 3—source data 1*). To assess the function of these genes, we implemented Gene Ontology (GO) and Kyoto Encyclopedia of Genes and Genomes (KEGG) pathway analysis. Downregulated genes were enriched for GO terms associated with blood, including 'blood microparticle', 'fibrinogen complex', and 'platelet alpha granule' (*Figure 3—source data 2*), and the 'complement and coagulation cascades' (*Figure 3—source data 3*) that play a role in vasculogenesis (*Girardi et al., 2006*; *Moser and Patterson, 2003*). Key genes within these groups included the complement inhibitors $Cd59a$ and complement component factor I ($Cfi$), and the secreted proteins fibrinogen alpha and gamma ($Fga$, $Fgg$), complement factor B ($Cfb$), protein C ($Proc$), and $Alpha$ $fetoprotein$ ($Afp$). We also observed a downregulation of $Jag2$ and $Slit1$ (*Figure 3—source data 1*), components of the Notch and Slit-Robo signaling pathways that regulate hematopoiesis and vasculogenesis (*Blockus and Chédotal, 2016*; *Kofler et al., 2011*).

The majority of downregulated factors were specifically expressed or enriched within the VE (84% of the 55 differentially expressed genes that were also detected by scRNA-sequencing of gastrulating mouse embryos *Nowotschin et al., 2019*; *Pijuan-Sala et al., 2019*; *Figure 3B*, *Figure 3—figure supplement 1A*). As these data were generated by whole embryo bulk RNA-sequencing, the downregulation of VE-associated genes could represent a reduction in the expression of specific factors or a relative decrease in the size of the VE. To distinguish between these possibilities, we assessed the expression levels of a number of other critical VE lineage determinants, including the transcription factors $Gata6$, $Gata4$, $Sox17$, and $Hnf4a$. We found that these genes were not significantly altered in $Rreb1^{-/-}$ (*Figure 3—figure supplement 1B*) suggesting that the downregulation, almost solely, of VE markers did not represent a global reduction in VE-associated transcription.

$Afp$, a plasma glycoprotein secreted by the yolk sac and fetal liver, which regulates angiogenesis (*Liang et al., 2004*; *Takahashi et al., 2004*), was one of the VE-associated genes significantly downregulated in mutant embryos. Thus, to validate our RNA-sequencing, we crossed $Rreb1^{+/-}$ mice to a transgenic reporter whereby the $Afp$ cis-regulatory elements drive GFP expression (*Kwon et al., 2006*). We then analyzed $Rreb1^{+/+}$ and $Rreb1^{-/-}$; $Afp$-GFP$^{Tg/+}$ embryos (*Figure 3—figure supplement 1C*). At E7.5 and 8.5, $Afp$-GFP is expressed by the embryonic and extraembryonic VE in wild-type embryos (*Kwon et al., 2006*; *Kwon et al., 2008*; *Figure 3—figure supplement 1D,E*). However, while $Rreb1^{-/-}$ mutant embryos expressed $Afp$-GFP within the embryonic VE and at the embryonic-extraembryonic boundary, they showed little to no $Afp$-GFP within the proximal extraembryonic VE (*Figure 3—figure supplement 1D,E*), consistent with the transcriptional downregulation of this gene.

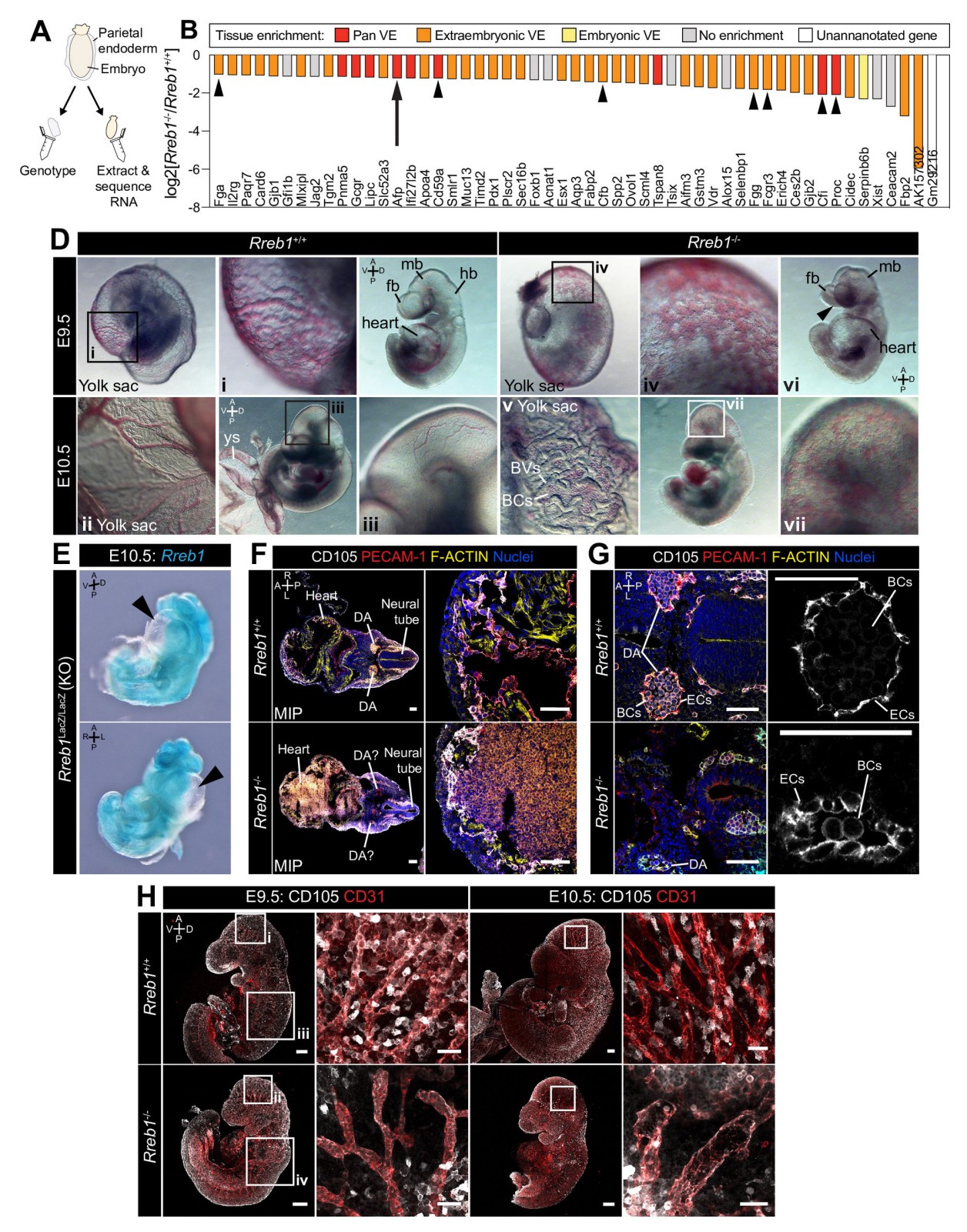

**Figure 3.** Loss of *Rreb1* causes cardiovascular defects in the early mouse embryo. (**A**) Schematic diagram depicting the sample collection methodology for whole embryo RNA-seq. Individual embryos were isolated from the uterus and the parietal endoderm dissected, lysed, and used for genotyping. The remaining part of the embryo was used for RNA extraction. Following genotyping, five individual wild-type and five individual mutant embryos were selected for sequencing. (**B**) Graph showing the list of significantly downregulated genes in *Rreb1*⁻/⁻ versus *Rreb1*⁺/⁺ embryos that were detected

*Figure 3 continued on next page*

*Figure 3 continued*

via single-cell sequencing in *Pijuan-Sala et al., 2019*. Each gene was manually categorized based on its enrichment in different tissues within this dataset. 'No enrichment' indicates genes that did not show a tissue-specific expression or enrichment. Arrow highlights *Afp* and arrowheads highlight genes associated with the complement and coagulation cascades. (D) Brightfield images of E9.5 and 10.5 embryos showing abnormal defects in the vasculature of *Rreb1*<sup>-/-</sup> embryos. In panel vi, arrowhead highlights the open anterior neural tube. (E) Confocal maximum intensity projections of whole E9.5 and 10.5 embryos (Sb, 200 μm) with adjacent high-magnification images of the cranial vasculature (Sb, 50 μm). Boxes i-iv in E9.5 are shown at higher magnification in *Figure 3—figure supplement 1J*. PECAM-1 marks vasculature. ENDOGLIN marks endothelial cells as well as hematopoietic, mesenchymal, and neural stem cells. To note, the tail of the lower right embryo was damaged during dissection. (F) Wholemount image of an E10.5 *Rreb1*<sup>LacZ/LacZ</sup> mutant embryo. Arrowhead highlights pericardial edema. A, anterior; P, posterior; Pr, proximal; Ds, distal; D, dorsal; V, ventral; L, left; R, right; ExVE, extraembryonic VE; EmVE, embryonic VE; ys, yolk sac; BCs, blood cells; cm; cephalic mesenchyme; ne, neurectoderm; EC, endothelial cells. The online version of this article includes the following source data and figure supplement(s) for figure 3:

**Source data 1.** List of genes that are differentially expressed between wild-type and Rreb1 mutant embryos.

**Source data 2.** Gene Ontology (GO) analysis of genes significantly upregulated and downregulated in E7.5 Rreb1 mutant embryos.

**Source data 3.** KEGG pathway analysis of genes significantly upregulated and downregulated in E7.5 Rreb1 mutant embryos.

**Figure supplement 1.** *Rreb1*<sup>-/-</sup> embryos exhibit cardiovascular defects.

As downregulated genes were functionally related to blood and vasculogenesis, we then asked whether vascular development was perturbed in the absence of *Rreb1* by performing immunofluorescence staining for FLK-1, a marker of hematopoietic and endothelial progenitors, and PECAM-1 (CD31), marking endothelial cell junctions. At E8.0, when a rudimentary circulatory system is first established, we discerned no difference in the level or pattern of FLK-1 or PECAM-1 expression. As previously described (*Yamaguchi et al., 1993*), and in both *Rreb1*<sup>+/+</sup> and *Rreb1*<sup>-/-</sup> embryos, FLK-1 +PECAM-1+ cells were present in the extraembryonic mesoderm, including the allantois and yolk sac blood islands, the cephalic mesenchyme, and heart (*Figure 3—figure supplement 1F,G*). At E9.5–10.5, while wild-type embryos formed a hierarchical branched network of blood vessels within the yolk sac and embryo-proper (*Figure 3D* i-iii), *Rreb1*<sup>-/-</sup> embryos had dysmorphic yolk sac capillaries that instead resembled a primitive capillary plexus (5/6 *Rreb1*<sup>-/-</sup> at E9.5, 4/6 *Rreb1*<sup>-/-</sup> at E10.5, *Figure 3D* iv, *Figure 3—figure supplement 1H* i, I) and blood within the extravascular space (*Figure 3D* v, *Figure 3—figure supplement 1H* ii). Cardiovascular defects were also observed within the *Rreb1*<sup>-/-</sup> embryo-proper including little to no blood within the fetus (2/6 *Rreb1*<sup>-/-</sup> at E9.5, 1/6 *Rreb1*<sup>-/-</sup> at E10.5, *Figure 3D* vi, *Figure 3—figure supplement 1I* iii), pooling of blood (2/6 *Rreb1*<sup>-/-</sup> at E9.5, 4/6 *Rreb1*<sup>-/-</sup> at E10.5), an enlarged heart (2/6 *Rreb1*<sup>-/-</sup> at E9.5, *Figure 3D* vi), and pericardial edema (2/8 *Rreb1*<sup>-/-</sup> at E10.5, *Figure 3E*). At E9.5, a number of *Rreb1*<sup>-/-</sup> embryos displayed extravascular FLK-1 +PECAM-1+ cells throughout the fetus, including accumulating within the heart (2/6 *Rreb1*<sup>-/-</sup> at E9.5, *Figure 3D* vii, F). These defects were also associated with an abnormal dorsal aorta morphology. In wild-type embryos, the dorsal aorta comprised flattened endothelial cells surrounding a lumen containing blood cells (*Figure 3F,G*). Elongated endothelial cells also bordered the neural tube (*Figure 3F,G*). In contrast, in *Rreb1*<sup>-/-</sup> embryos, the dorsal aorta lumen was collapsed and FLK-1 +PECAM-1+ cells, both of the aorta and surrounding the neural tube, appeared rounded compared to those in wild-type embryos (*Figure 3G*).

While the dorsal midbrain of E9.5 and 10.5 wild-type embryos contained defined networks of veins and branched blood vessels (*Figure 3D* iii, H, *Figure 3—figure supplement 1J*), *Rreb1*<sup>-/-</sup> exhibited displayed reduced vascular remodeling with fewer, wider blood vessels (3/6 *Rreb1*<sup>-/-</sup> at E10.5, *Figure 3H*, *Figure 3—figure supplement 1J*). ENDOGLIN (CD105) is also a marker of endothelial cells. In E9.5 wild-type embryos, a thin layer of PECAM-1 +ENDOGLIN + endothelial cells formed between the neuroepithelium and outer surface ectoderm layer. However, in *Rreb1*<sup>-/-</sup>, PECAM-1 +ENDOGLIN + cells were arranged into discrete, rounded clusters (*Figure 3—figure supplement 1K*). Thus, loss of *Rreb1* results in the downregulation of VE-expressed vasculogenesis-associated genes and cardiovascular defects that culminate in embryonic lethality at midgestation.

## *Rreb1* regulates cytoskeleton and adherens junction organization within the epiblast

We then analyzed genes that were upregulated in E7.5 *Rreb1*<sup>-/-</sup> vs. *Rreb1*<sup>+/+</sup> embryos. The upregulated genes, *Bmper* and *Cxcl10* (*Figure 3—source data 1*), are associated with 'endothelial cell activation' and thus their altered expression may contribute to the observed cardiovascular defects. H2-

Q2, the purported mouse homolog of HLA-G (*Comiskey et al., 2003*; *Figure 4—figure supplement 1A*), was also upregulated consistent with reports that Rreb1 is a transcriptional repressor or HLA-G in humans (*Flajollet et al., 2009*). Subsequent GO analysis revealed that upregulated genes were enriched for four main categories; 'cytoskeleton', 'membrane and vesicle trafficking', 'cell junctions', and 'extracellular space' (*Figure 3—source data 2*). Factors associated with the cytoskeleton included microtubule components (*Tubb3*), microtubule-interacting proteins (*Map6, Jakmip2, Fsd1*), microtubule motors (*Kif5a, Kif5c, Kif12*), actin-binding proteins (*Coro1a*), and factors that connect adherens junctions to the cytoskeleton (*Ctnna2, Ablim3*) (*Figure 4A*). Genes within the 'vesicle trafficking' category were also related to the cytoskeleton. For example, Rab family members (*Rab6b, Rab39b*) (*Figure 4A*) regulate vesicle transport along actin and microtubule networks.

We therefore asked whether these transcriptional changes in cytoskeleton genes corresponded to a change in cytoskeletal organization in *Rreb1* mutants. In the normal (wild-type, *Rreb1*$^{+/+}$) epiblast epithelium, F-ACTIN was arranged in linear filaments oriented parallel to cell junctions (*Figure 4B,C*). In contrast, F-ACTIN was punctate at epiblast cell junctions within *Rreb1*$^{-/-}$ embryos (*Figure 4B,C*). The cytoskeleton interacts with and influences the localization of adherens junction components (*Chen et al., 2003*; *Liang et al., 2015*; *Mary et al., 2002*; *Mège and Ishiyama, 2017*; *Sako-Kubota et al., 2014*; *Stehbens et al., 2006*; *Teng et al., 2005*). As we noted a significant upregulation of *Ctnna2* and *Ablim3*, which encode proteins that connect the cytoskeleton to adherens junctions (*Figure 4A*), we asked whether the change in F-ACTIN was associated with a rearrangement of cell junctions. Cadherins are critical components of adherens junctions and, during gastrulation, E-CADHERIN is expressed within the epiblast, VE, and extraembryonic ectoderm (*Pijuan-Sala et al., 2019*). In wild-type embryos, E-CADHERIN, similar to F-ACTIN, formed a continuous belt between epithelial epiblast cells but, in *Rreb1* mutants, showed a punctate localization (*Figure 4D,E*, *Figure 4—figure supplement 1B–D*). Beta-CATENIN was also more punctate at *Rreb1* mutant compared to wild-type epiblast junctions (*Figure 4—figure supplement 1E*), indicating that adherens junction complexes were altered. The punctate adherens junction organization was observed within the proximal but not distal *Rreb1*$^{-/-}$ epiblast (*Figure 4—figure supplement 1F*). The change in E-CADHERIN and Beta-CATENIN localization was not associated with a transcriptional change in these genes, or in the expression of other adhesion-associated factors, such as tight junction components (*Figure 4—figure supplement 1G*). Thus, the altered protein localization was due to post-transcriptional mechanisms. Therefore, loss of *Rreb1* results in a change in the expression of cytoskeleton-associated factors and a change in the organization of the cytoskeleton and adherens junctions within the epiblast.

## *Rreb1* maintains epithelial architecture of embryonic and extraembryonic tissues

The cytoskeleton is the scaffold of the cell that regulates cell-cell adhesion (*Elson, 1988*; *Gavara and Chadwick, 2016*; *Grady et al., 2016*; *Ketene et al., 2012*) and epithelial organization (*Bachir et al., 2017*; *Ivanov et al., 2010*; *Sun et al., 2015*; *Vasileva and Citi, 2018*). In cancer, a cytoskeleton-mediated switch from linear to punctate E-CADHERIN results in weaker cell-cell adhesion and loss of epithelial integrity (*Aiello et al., 2018*; *Ayollo et al., 2009*; *Gloushankova et al., 2017*; *Jolly et al., 2015*; *Kovac et al., 2018*; *Saitoh, 2018*). In keeping with this, *Rreb1*$^{-/-}$ gastrulating embryos exhibited perturbed epithelial architecture. In wild-type embryos, VE cells formed an ordered monolayer overlying the embryonic epiblast and the ExE (*Figure 5A,B*, *Figure 5—figure supplement 1A*), while in *Rreb1*$^{-/-}$, cells protruded from the VE (*Figure 5A*), the extraembryonic VE was frequently ruffled (*Figure 5B*, *Figure 5—figure supplement 1A*), and abnormal masses of cells accumulated at the anterior embryonic-extraembryonic boundary (*Figure 5C*, *Figure 5—figure supplement 1B*).

*Rreb1*$^{-/-}$ initiated gastrulation in the posterior of the embryo, as marked by downregulation of the pluripotency-associated transcription factor SOX2 and upregulation of the primitive streak marker BRACHYURY (*Figure 5D*). Furthermore, *Rreb1*$^{-/-}$ epiblast cells underwent an EMT at the primitive streak, delaminated from the epithelium, and migrated anteriorly in the wings of mesoderm (*Figure 5E*). Cells within *Rreb1*$^{-/-}$ embryos also differentiated into mesoderm and DE, marked by GATA6 and SOX17 expression respectively (*Figure 5D*, *Figure 5—figure supplement 1C*). Hence, *Rreb1*$^{-/-}$ mutant cells specify and begin to pattern the embryonic germ layers. As the majority of downregulated genes were factors expressed by the endoderm, we investigated endoderm

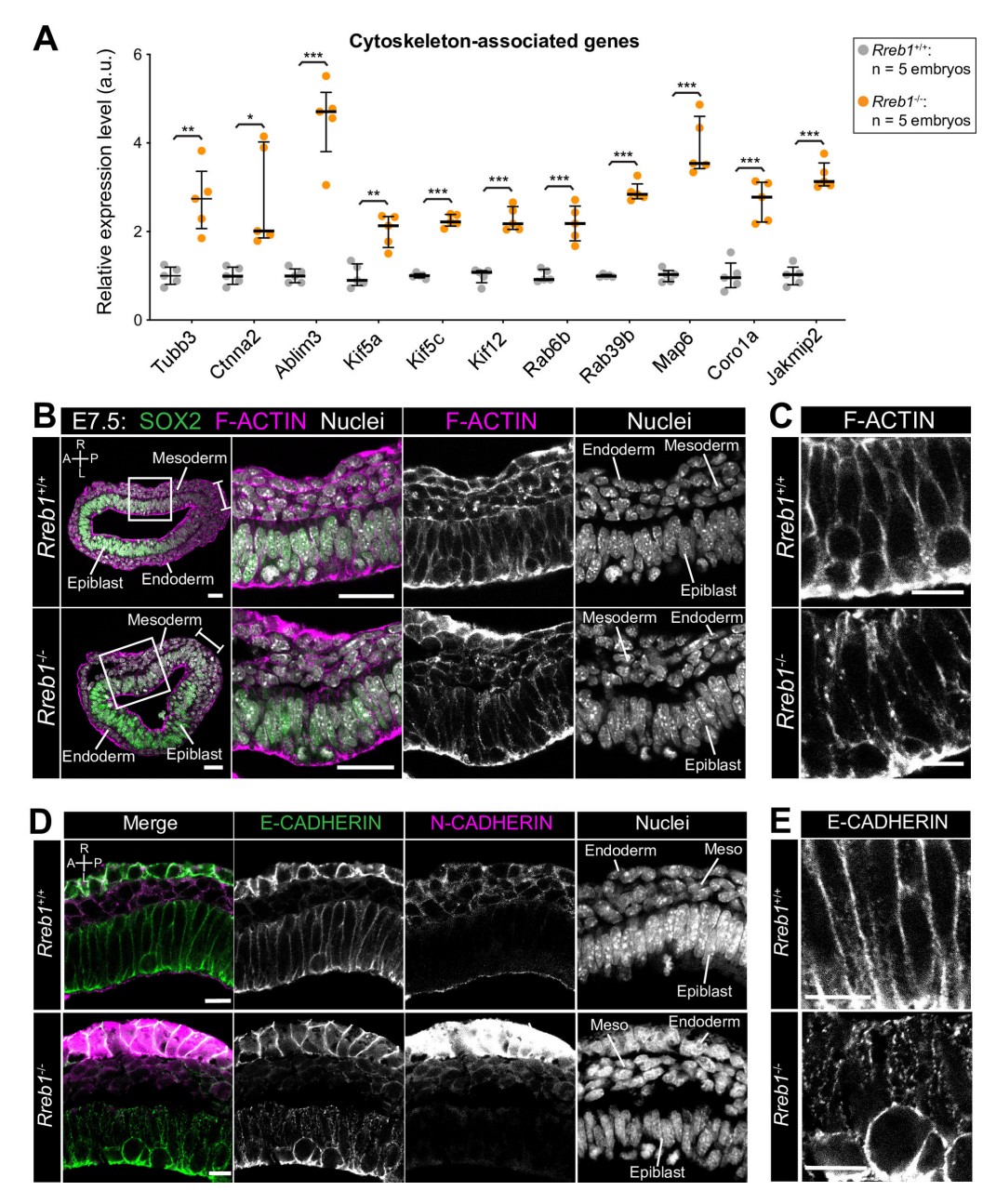

**Figure 4.** The *Rreb1*⁻/⁻ epiblast shows altered cytoskeleton and adherens junction organization. (A) Graph showing the relative expression level of cytoskeleton-associated genes from RNA-sequencing of individual *Rreb1*⁺/⁺ and *Rreb1*⁻/⁻ embryos. Each point represents a single embryo. Statistical analysis was performed using an Unpaired *t*-test (*p<0.05, **p<0.005, ***p<0.001). Bars represent median and IQR. Expression is shown relative to the mean expression in wild-type embryos. (B-E) Confocal optical sections showing transverse cryosections of immunostained *Rreb1*⁺/⁺ and *Rreb1*⁻/⁻ embryos. Boxes indicate lateral epiblast regions shown at higher magnification in adjacent panels. *Rreb1*⁻/⁻ embryos exhibit a punctate localization of E-CADHERIN (n = 4/4 embryos). Sb, 10 μm. (C,E) Highest magnification images showing a small region of the epiblast epithelium. Sb, 10 μm. Brackets mark the primitive streak. A, anterior; P, posterior; L, left; R, right.

The online version of this article includes the following figure supplement(s) for figure 4:

**Figure supplement 1.** Loss of *Rreb1* alters epiblast adherens junction organization.

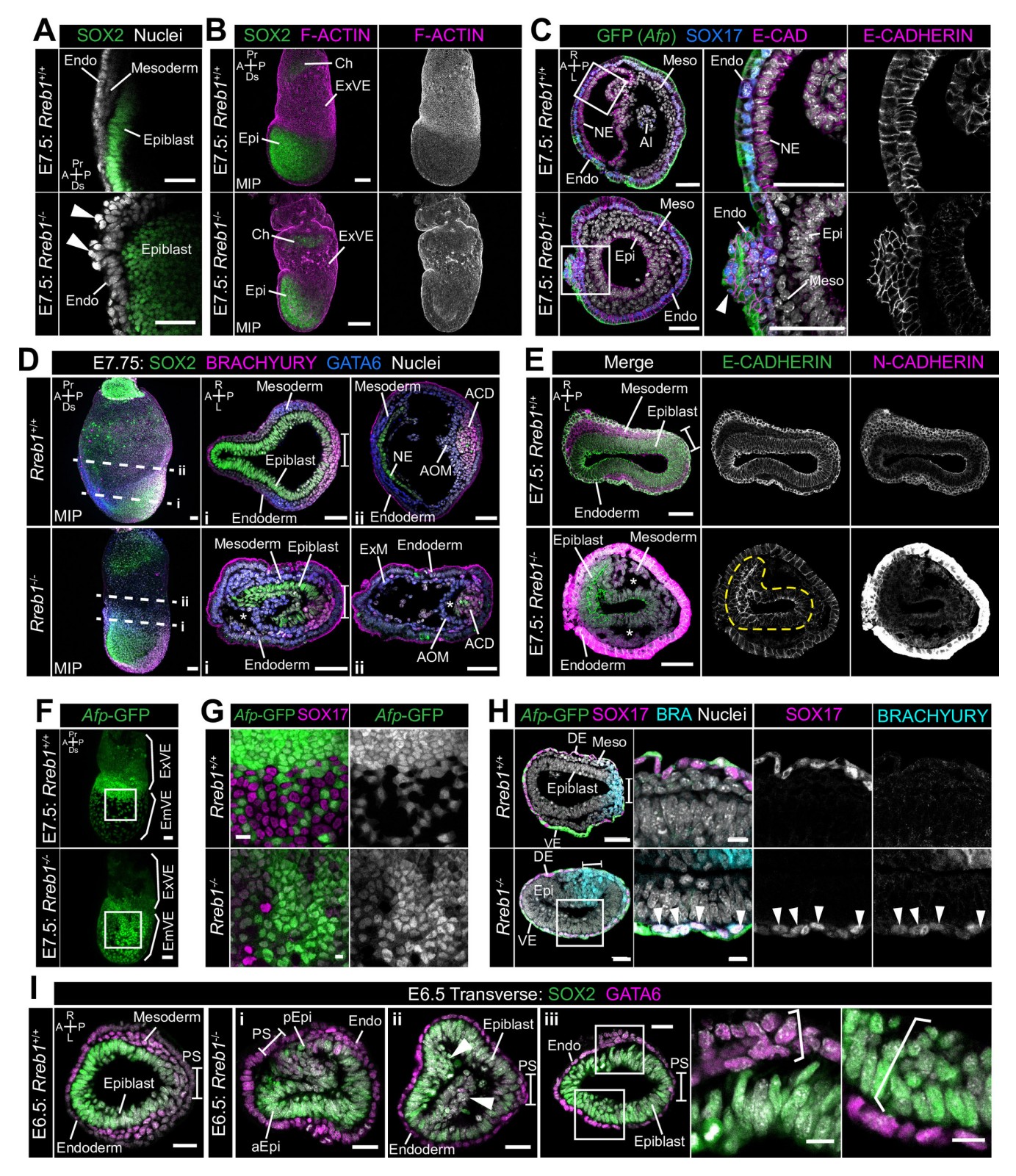

**Figure 5.** *Rreb1* maintains epithelial organization in the early mouse embryo. (**A**) Sagittal confocal optical section of the anterior of E7.5 *Rreb1* wild-type and homozygous mutant embryos. Arrowheads highlight cells abnormally protruding from the VE overlying the epiblast. Sb, 25 μm. (**B**) Confocal maximum intensity projections (MIP) of immunostained E7.5 embryos showing ruffling of the extraembryonic VE (n = 7/8 E7.5 *Rreb1*[-/-] embryos exhibit endoderm ruffling). Sb, 100 μm. (**C**) Confocal optical sections showing transverse cryosections of E7.5 *Afp*-GFP *Rreb1* wild-type and homozygous

*Figure 5 continued on next page*

*Figure 5 continued*

mutant embryos. Boxes indicate regions shown in higher magnification in adjacent panels. Arrowhead indicates abnormal accumulation of *Afp* + VE cells and underlying *Afp*- DE cells at the anterior embryonic-extraembryonic boundary in *Rreb1*⁻/⁻ (n = 7/52 E7.5 *Rreb1*⁻/⁻ embryos exhibit anterior endoderm accumulations). Sb, 50 μm. (D, E) Maximum intensity projections (MIPs) of wholemount E7.5 embryos and confocal optical sections of transverse cryosections. (D) Dashed lines mark approximate plane of section. Sb, 50 μm. (E) Dashed yellow line outlines the epiblast. Sb, 50 μm. Asterisks mark abnormal gaps between tissue layers, which was the most common defect observed (n = 38/52 E7.5 *Rreb1*⁻/⁻). (F) Representative images of *Rreb1*⁺/⁺ and *Rreb1*⁻/⁻ embryos highlighting the epithelial defects observed: (i) abnormal accumulations of cells in the epiblast, (ii) epiblast folding (n = 8/52 52 E7.5 *Rreb1*⁻/⁻ embryos exhibit abnormal epiblast folding), in this case the epiblast is folded such that the putative anterior (aEpi) and posterior (pEpi) regions are adjacent to one another, (iii) formation of multilayered regions (highlighted with brackets) in the, typically monolayer, endoderm and epiblast. Sb 25 μm, high mag sb, 10 μm. (G–I) Confocal MIPs (G,H) and confocal optical sections showing transverse cryosections of *Afp*-GFP; *Rreb1*⁺/⁺ and *Rreb1*⁻/⁻ embryos (I). Boxes indicate region shown in higher magnification in H. Sb, 50 μm. Pr, proximal; Ds, distal; A, anterior; P, posterior; R, right; L, left; Epi, epiblast; aEpi, anterior epiblast; pEpi, posterior epiblast; PS, primitive streak; Endo, endoderm; ACD, allantois core domain; AOM, allantois outer mesenchyme; Ch, chorion; Meso, mesoderm; ExVE, extraembryonic visceral endoderm; EmVE, embryonic visceral endoderm; DE, definitive endoderm; NE, neurectoderm; Al, allantois.

The online version of this article includes the following figure supplement(s) for figure 5:

**Figure supplement 1.** *Rreb1* mutant embryos have perturbed epithelial architecture.

**Figure supplement 2.** Severity of *Rreb1* mutant phenotypes is dependent on genetic background.

specification and morphogenesis in more detail. We observed high levels of non-specific antibody staining, including for BRACHYURY and N-CADHERIN in the *Rreb1*⁻/⁻ VE. Such non-specific staining is often observed in the VE of wild-type embryos prior to intercalation of the DE (*Kwon et al., 2008*; *Morgani et al., 2018a*), which has ben attributed to its extensive vacuolation. Thus, we hypothesized that there may be defects in DE intercalation in mutant embryos. Consistent with this, the *Afp*-GFP reporter mouse line revealed delayed dispersal of the embryonic VE, a process driven by DE intercalation. By E7.5, the VE was fully dispersed in wild-type embryos, characterized by mosaic GFP labeling of the outer endoderm layer (*Figure 5F,G*), but noticeably reduced in *Rreb1*⁻/⁻ embryos of the same stage (*Figure 5F,G*). The VE was successfully dispersed in mutant embryos by E8.5 (*Figure 3—figure supplement 1E*). Wild-type SOX17 +DE cells intercalated into the lateral VE as they migrated anteriorly, but *Rreb1*⁻/⁻ SOX17 + cells migrated to the anterior pole without intercalating (*Figure 5H*, *Figure 5—figure supplement 1C*), contributing to the abnormal bulges at the anterior embryonic-extraembryonic boundary (*Figure 5C*, *Figure 5—figure supplement 1B*). Notably, unlike wild-type DE, *Rreb1*⁻/⁻ SOX17 + cells expressed the mesoderm and primitive streak marker BRACHYURY (*Figure 5H*), suggesting that they may be incorrectly specified.

As within the VE, the epiblast of mutants showed a range of morphological defects including uncharacteristic folding of the epithelial layer (*Figure 5E*, I i, *Figure 5—figure supplement 1D*), abnormal accumulations of cells (*Figure 5I* ii), increasingly variable cell orientation (*Figure 5—figure supplement 1E–G*), separation of typically closely apposed tissue layers, such as the mesoderm and endoderm (*Figure 5D,E*, *Figure 5—figure supplement 1H*), and cells falling out of the epiblast (*Figure 5—figure supplement 1I*). In wild-type embryos, epiblast cells divide at the apical, cavity-facing surface while being maintained within the epithelial layer but, in *Rreb1*⁻/⁻ embryos, we observed dividing cells that left the epithelium (*Figure 5—figure supplement 1J*). Additionally, the epiblast and endoderm are monolayer epithelia in wild-type embryos but formed multilayered regions in *Rreb1*⁻/⁻ mutants (*Figure 5I* iii).

Epithelial homeostasis requires tight regulation of proliferation and the maintenance of cell polarity. However, *Rreb1*⁻/⁻ embryos showed no difference in the absolute or relative number of dividing cells within the epiblast, VE, or mesoderm when compared to wild-type littermates (*Figure 5—figure supplement 1K,L*). Furthermore, apicobasal polarity of the *Rreb1*⁻/⁻ epiblast cells was unaffected, demonstrated by the correct positioning of the tight junction protein ZO-1 at the apical surface and the basement membrane protein LAMININ at the basal surface (*Figure 5—figure supplement 1M,N*). Together these data show that loss of *Rreb1* results in disrupted epithelial architecture of both embryonic and extraembryonic tissues, associated with altered cytoskeleton and adherens junction organization.

## Penetrance and expressivity of the *Rreb1*<sup>-/-</sup> phenotype is genetic background-dependent

Chimeras containing *Rreb1*<sup>-/-</sup> cells showed an accumulation of cells at the primitive streak (*Su et al., 2020*). However, this phenotype was observed in only a small fraction of in *Rreb1*<sup>-/-</sup> embryos (3/52). The difference in phenotypic penetrance and expressivity between these experiments could exist for a variety of reasons, including contribution of the extraembryonic tissues, which are wild-type in chimeras but mutant in the *Rreb1*<sup>-/-</sup> mouse line, variability in the proportion of wild-type versus mutant cells and interactions between wild-type and mutant cells in chimeras, as well as genetic background. While the majority of *Rreb1* mutant embryos analyzed in this study were of a CD1 outbred background, reflecting the genetic diversity within the human population, chimeric embryos were generated by introducing 129 ESCs into C57BL/6 embryos, both of inbred backgrounds. Mutant inbred mice tend to display more severe defects than their outbred counterparts and thus phenotypic differences could reflect genetic background. To assess this, we collected and analyzed a litter of E7.5 C57BL/6 *Rreb1*<sup>-/-</sup> embryos. We found that 4/4 C57BL/6 *Rreb1*<sup>-/-</sup> embryos exhibited severe defects in the exit of cells from the posterior epiblast (*Figure 5—figure supplement 2A,B*) and a reduction in LAMININ basement membrane break down at the primitive streak compared to wild-type *Rreb1*<sup>+/+</sup> littermates (*Figure 5—figure supplement 2C,D*). There was also a more pronounced buckling of the epiblast epithelium than in outbred CD1 embryos, with 4/4 embryos displaying abnormal epiblast folding (*Figure 5—figure supplement 2B*). Thus, penetrance and expressivity of the *Rreb1*<sup>-/-</sup> phenotype is influenced by genetic background.

## *Rreb1* mutant embryos display invasive phenotypes

In the context of cancer, cells with punctate E-CADHERIN localization represent an intermediate epithelial-mesenchymal state (*Sha et al., 2019*; *Yang et al., 2020*), characterized by an increased propensity for collective invasion and metastasis (*Aiello et al., 2018*; *Ayollo et al., 2009*; *Gloushankova et al., 2017*; *Jolly et al., 2015*; *Kovac et al., 2018*; *Saitoh, 2018*). This state is linked to the downregulation of the transcription factor *Ovol1* (*Jia et al., 2015*; *Saxena et al., 2020*), which suppresses a mesenchymal identity, and the tight junction component, *Claudin7* (*Aiello et al., 2018*; *Kim et al., 2019*; *Wang et al., 2018*). Both of these factors were also significantly downregulated in *Rreb1*<sup>-/-</sup> embryos (*Figure 6—figure supplement 1A*). Furthermore, we observed *Rreb1*<sup>-/-</sup> epiblast cells that had acquired mesenchymal characteristics (n = 1/3 *Rreb1*<sup>-/-</sup>). In wild-type embryos, the mesenchymal marker and EMT regulator SNAIL was expressed within the primitive streak and the wings of mesoderm (*Figure 6A*). However, in *Rreb1*<sup>-/-</sup>, SNAIL was ectopically expressed within epiblast cells precociously exiting the epithelium (*Figure 6A*). Moreover, these cells exhibited punctate Beta-CATENIN, in contrast to the linear localization in neighboring SNAIL-negative epiblast (*Figure 6—figure supplement 1B*). Moreover, in 3/52 *Rreb1*<sup>-/-</sup> embryos, we observed chains of cells that traversed tissue layers, including cells expressing the epiblast marker SOX2 that crossed the VE (*Figure 6B*, *Figure 6—figure supplement 1*) and cells expressing the mesoderm and endoderm marker GATA6 that spanned the epiblast (*Figure 6—figure supplement 1C*). In most cases, these aberrant cells crossed the VE and SOX2-positive (SOX2+) pyknotic nuclei were detected on the adjacent exterior surface of the embryo (*Figure 6—figure supplement 1C,D*).

These events were rare in the *Rreb1*<sup>-/-</sup> mouse line, precluding a detailed analysis of the identity of aberrant cell populations. However, we frequently observed ectopic SOX2 + cells dispersed throughout chimeric embryos, where the embryonic epiblast-derived tissues are a mosaic of wild-type and mutant origin and extraembryonic tissues are wild-type (30/63, 48% of *Rreb1*<sup>-/-</sup> chimeric embryos, *Figure 6C–E*, *Figure 6—figure supplement 1E* and 33–90 ectopic SOX2 + cells/per embryo). Abnormal SOX2 + cells were predominantly positioned between the epiblast and endoderm tissue layers and less commonly found within the epiblast, cavity, and wings of mesoderm (*Figure 6—figure supplement 1F*). These cells divided and persisted until later stages of development (*Figure 6—figure supplement 1G*) and were also observed at the onset of gastrulation (*Figure 6—figure supplement 1H*).

In chimeric embryos, ectopic cells expressed higher levels of SOX2 than those within the epiblast (*Figure 6—figure supplement 1I*), suggesting that their identity may be altered during exit from the epithelium. To investigate this, we performed immunofluorescence analysis of a panel of key markers expressed within embryos at this time. In addition to SOX2, ectopic cells expressed the

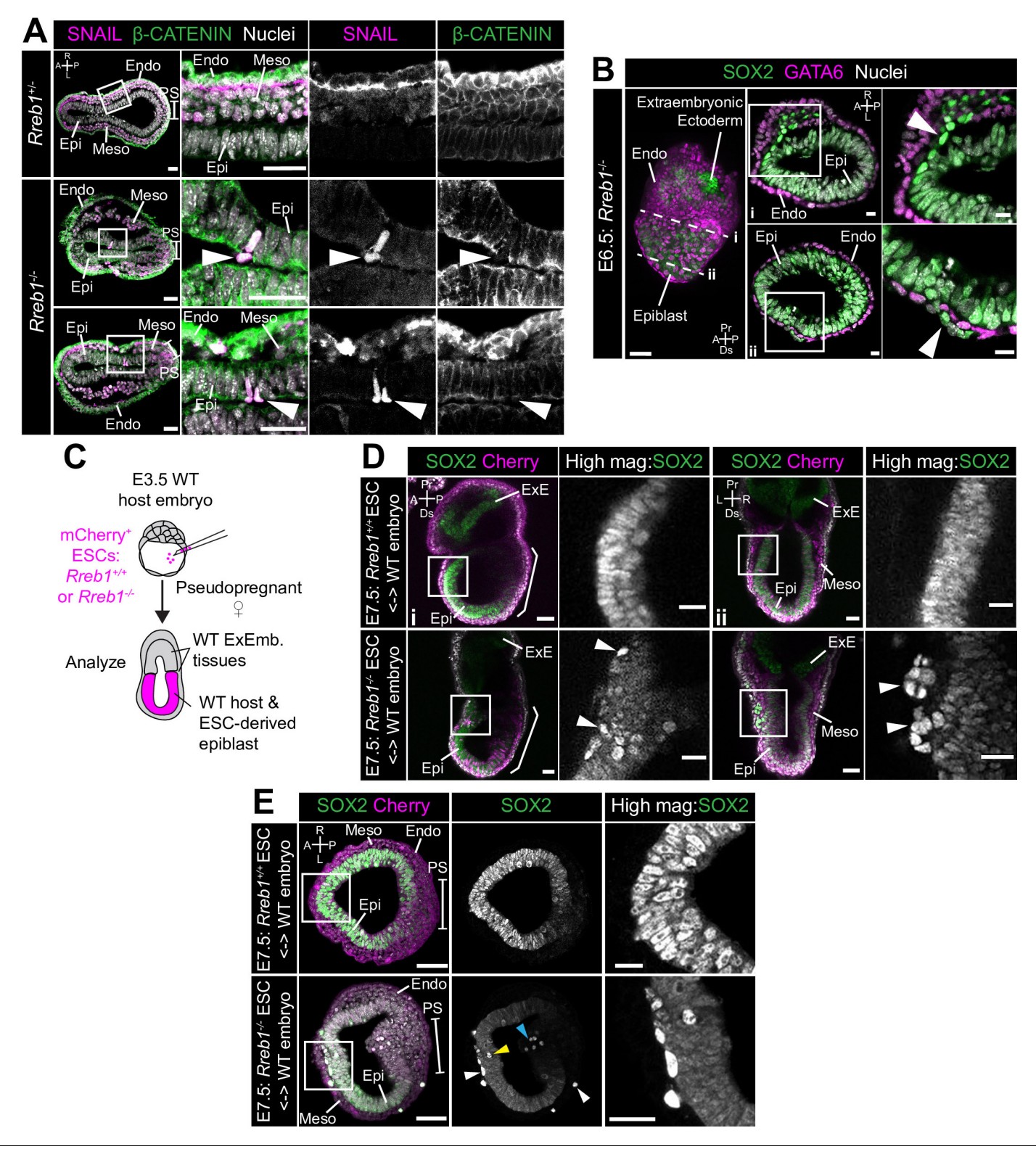

**Figure 6.** Loss of Rreb1 results in invasive cell behaviors. (A) Confocal optical sections of transverse cryosections of immunostained E7.5 embryos. Boxes show regions displayed at higher magnification in adjacent panels. Arrowheads indicate ectopic SNAIL expression in epiblast cells exiting the epithelium. Sb, 25 μm. (B) Confocal optical sections of maximum intensity projection (MIP, Sb, 50 μm) and transverse cryosections of immunostained E6.5 *Rreb1⁻/⁻* embryos. Dashed lines mark approximate plane of transverse section. Arrowhead marks ectopic SOX2 + cells leaving the epiblast and traversing the outer endoderm layer. Sb, 10 μm. (C) Schematic diagram illustrating how chimeras were generated. *Rreb1⁺/⁺* or *Rreb1⁻/⁻* embryonic stem

*Figure 6 continued on next page*

*Figure 6 continued*

cells (ESCs) constitutively expressing an mCherry lineage label were injected into wild host E3.5 embryos. Embryos were then transferred to pseudopregnant host females and dissected for analysis at later developmental stages. (D,E) Sagittal (D i), lateral (D ii) and transverse (E) confocal optical sections of immunostained E7.5 chimeric embryos containing either *Rreb1*[+/+] or *Rreb1*[-/-] cells. Arrowheads mark abnormal SOX2 + cells, expressing higher levels of SOX2 than their neighbors, in the epiblast (yellow), primitive streak (blue arrowhead) or between the epiblast and visceral endoderm layers (white). Sb, 50 μm. High-magnification inset Sb, 25 μm. A, anterior; P, posterior; L, left; R, right; Endo, endoderm; Meso, mesoderm; Epi, epiblast; PS, primitive streak.

The online version of this article includes the following figure supplement(s) for figure 6:

**Figure supplement 1.** Loss of *Rreb1* promotes invasive cell behaviors.

**Figure supplement 2.** Ectopic cells in *Rreb1* chimeras have a PGC-like marker profile.

pluripotency-associated factors NANOG and OCT4 (POU5F1) (*Figure 6—figure supplement 2A*). However, they did not express the epiblast marker OTX2 (*Figure 6—figure supplement 2B*). OTX2 is additionally expressed within the mesoderm and VE, hence its absence excluded the possibility that cells transdifferentiated toward these lineages. OTX2 blocks pluripotent cells from adopting a primordial germ cell (PGC) identity (*Zhang et al., 2018*). Therefore, we asked whether the absence of OTX2 correlated with an upregulation of PGC-associated genes. PGCs express a myriad of pluri-potency factors, for example SOX2, NANOG, and OCT4, but not the naïve pluripotency marker KLF4. Ectopic SOX2 + cells did not express KLF4 (*Figure 6—figure supplement 2C*) but did express the PGC marker AP2γ (*Figure 6—figure supplement 2D*). In summary, *Rreb1*[-/-] cells that ectopically exited the epiblast in chimeric embryos were SOX2[HI]NANOG + OCT4+OTX2- KLF4- AP2γ+, a pro-file found only in PGCs at this developmental stage. Thus, loss of *Rreb1* caused cells to ectopically exit the epiblast epithelium in early mouse embryos, correlating with a change in cell fate.

## Invasive cells in *Rreb1*[-/-] chimeras are associated with a distinct ECM organization

In chimeric embryos, ectopic SOX2 + cells were of both wild-type and mutant origin (*Figure 7A*, *Figure 7—figure supplement 1A*), indicating that invasive-like behaviors were not driven solely by cell-autonomous properties, such as changes in the cytoskeleton and adherens junctions. Remodeling of the extracellular matrix (ECM) could promote invasive behaviors of both wild-type and mutant cells. We noted that many of the genes that were significantly altered in *Rreb1*[-/-] embryos were associated with ECM and cell-ECM adhesion. For example, *Tff3* (*Ahmed et al., 2012*; *Pandey et al., 2014*), *Hpse* (*Liu et al., 2019*), *Slit1* (*Gara et al., 2015*), Spon1 (*Chang et al., 2015*), *Spock1*, and *Spock3* (*Chen et al., 2016*) are associated with increased cancer cell invasion and were upregulated in *Rreb1*[-/-], and *Selenbp1* (*Caswell et al., 2018*; *Schott et al., 2018*) and *Serpin6b* (*Chou et al., 2012*) are tumor suppressor genes that were downregulated. Therefore, we asked whether the basement membrane underlying the epiblast was perturbed in *Rreb1*[-/-] chimeras.

In wild-type chimeras, the basement membrane at the epiblast-VE interface is broken down in the posterior of the embryo at the primitive streak during gastrulation, as cells undergo an EMT (*Figure 7C*). In *Rreb1*[-/-] embryo chimeras, the basement membrane was broken down at the primi-tive streak but also in anterior and lateral regions of the epiblast (*Figure 7C*, *Figure 7—figure supplement 1B*). SOX2 + cells were observed traversing these ectopic basement membrane breaks (*Figure 7C*). Furthermore, aberrant SOX2 + cells were surrounded by higher levels of Laminin than their neighbors and associated with Laminin tracks, up to 68 μm (approximately 7 cell diameters) in length (*Figure 7D*, *Figure 7—figure supplement 1C*). Thus, loss of *Rreb1* in the mouse embryo caused epiblast epithelial cells to cross the basement membrane underlying the epiblast epithelium, reminiscent of the invasive cell behaviors observed in cancer metastasis. These defects were associ-ated with cell-autonomous changes in the cytoskeleton as well as non-cell-autonomous changes in the ECM. KEGG pathway analysis also revealed that the genes upregulated in *Rreb1*[-/-] embryos were enriched for pathways associated with cancer, including 'Pathways in cancer', 'MicroRNAs in cancer', and 'Gastric cancer' (3/5 most enriched pathways, *Figure 7E*). Together these data suggest that the embryonic role of *Rreb1* may be functionally linked to its role in cancer (*Figure 7F*).

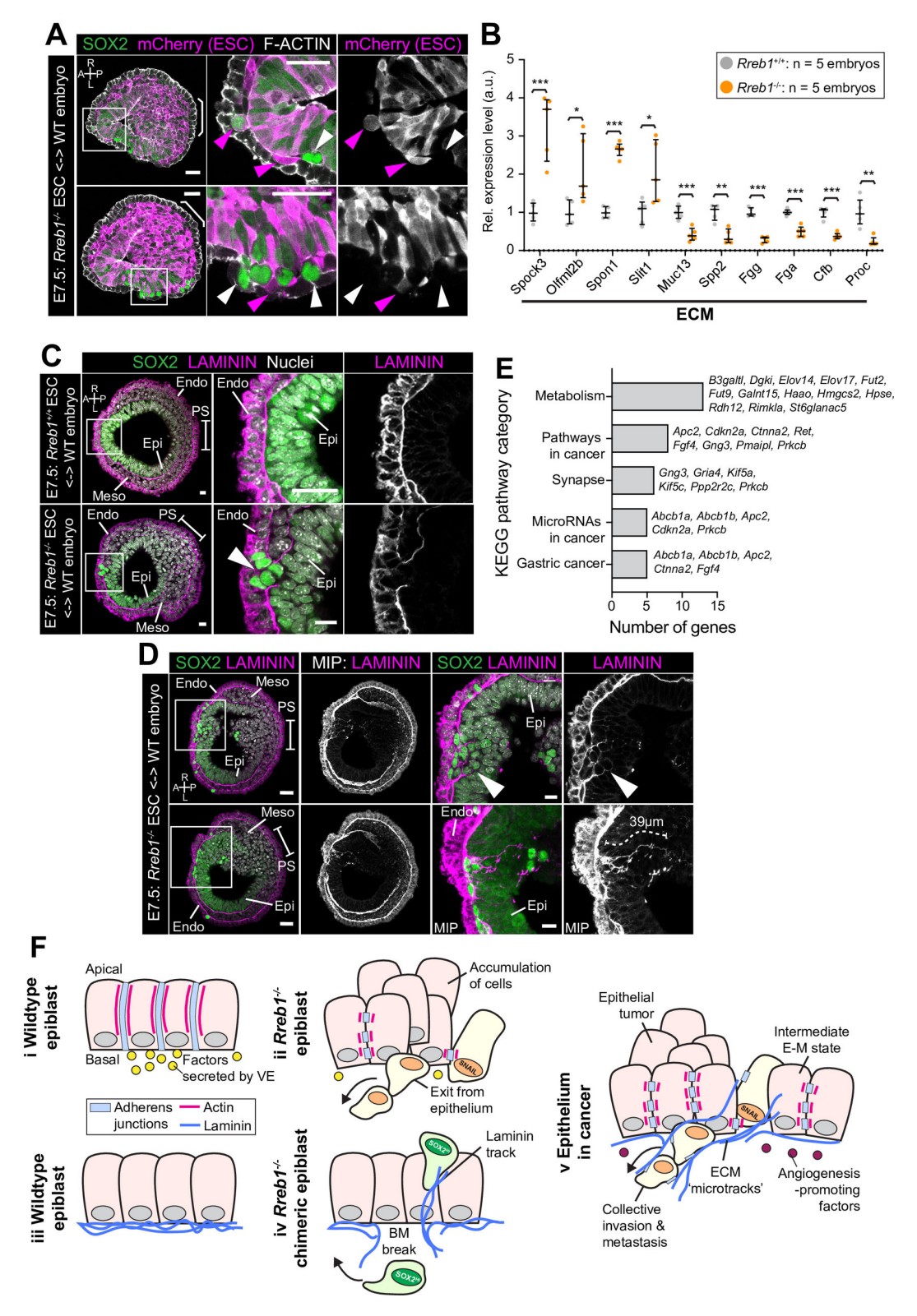

**Figure 7.** *Rreb1⁻ᐟ⁻* chimeras exhibit changes in ECM organization. (**A,C,D**) Confocal images showing transverse cryosections of immunostained E7.5 chimeric embryos containing *Rreb1⁺ᐟ⁺* or *Rreb1⁻ᐟ⁻* cells. (**A**) Confocal optical sections of *Rreb1⁻ᐟ⁻* chimeras. Cherry fluorescence is a constitutive lineage label marking the progeny of *Rreb1⁻ᐟ⁻* embryonic stem cells (ESCs) introduced into host embryos. Arrowheads mark ectopic SOX2 + cells derived from wild-type host cells (white) or from *Rreb1⁻ᐟ⁻* cells (magenta). Sb, 25 µm. (**C**) Confocal optical sections of *Rreb1⁻ᐟ⁻* chimeras. Arrowhead marks ectopic

*Figure 7 continued on next page*

*Figure 7 continued*

SOX2 + cells traversing a break in the basement membrane between the epiblast and outer visceral endoderm layer. Sb, 10 µm. (D) Confocal optical sections and maximum intensity projections (MIP) of *Rreb1*$^{-/-}$ chimeras. Upper and lower panels are sections taken from the same embryo, 20 µm apart. Arrowheads mark invasive SOX2 + cells surrounded by Laminin. Dashed line marks the approximate line of measurement of the length of the adjacent Laminin track. Sb, 25 µm and 10 µm for high-magnification image. (B) Graph showing the relative expression level of a panel of ECM- associated genes from RNA-sequencing of individual *Rreb1*$^{+/+}$ and *Rreb1*$^{-/-}$ embryos. Each point represents a single embryo. Statistical analysis was performed using an Unpaired *t*-test (*p<0.05, **p<0.005, ***p<0.001). Bars represent median and IQR. Expression is shown relative to the mean expression in wild-type embryos. (E) Graph showing the top five results from KEGG pathway analysis of genes that were significantly upregulated in *Rreb1*$^{-/-}$ versus *Rreb1*$^{+/+}$ embryos. The genes associated with each category are shown on the graph. (F) Schematic diagram summarizing some of the key findings in this paper. i. In the wild-type epiblast epithelium of the mouse embryo, adherens junction components, such as E-CADHERIN, form continuous belts along cell junctions and F-ACTIN forms linear filaments that run parallel to these junctions. ii. In *Rreb1*$^{-/-}$ embryos, there was a reduction in the expression of a cohort of factors secreted by the VE, which may alter the behavior of epiblast cells. Furthermore, we observed various phenotypes in the *Rreb1*$^{-/-}$ epiblast epithelium including a more variable cell orientation compared to that of wild-type embryos, abnormal accumulations of cells, ectopic expression of the mesenchymal marker SNAIL, and chains of cells apparently exiting the epithelial layer. iii. The wild-type epiblast epithelium forms a Laminin basement membrane at its basal surface. iv. In contrast, in chimeric embryos that contain a mix of both wild-type and *Rreb1*$^{-/-}$ cells, we observed cells of both genotypes traversing breaks in the underlying basement membrane which were then found ectopically throughout the embryo. Moreover, we observed the formation of long Laminin tracks closely associated with abnormal SOX2$^{HI}$ cells. v. The cell behaviors observed in *Rreb1*$^{-/-}$ embryos and chimeras are similar to those observed in cancer. For example, abnormal accumulations of epithelial cells are the basis of tumor formation, changes in cytoskeleton organization combined with a switch from linear to punctate E-CADHERIN and ectopic expression of mesenchymal markers characterizes an intermediate EMT state that is associated with collective invasion during cancer metastasis. Remodeling of the ECM into parallel fibers, known as ECM microtracks, facilitates collective cell invasion in cancer metastasis. Furthermore, the tumor microenvironment commonly show a change in the expression of secreted factors that promote angiogenesis. A, anterior; P, posterior; L, left; R, right; Pr, proximal; Ds, distal; Epi, epiblast; Endo, endoderm; ExE, extraembryonic ectoderm; Meso, mesoderm.

The online version of this article includes the following figure supplement(s) for figure 7:

**Figure supplement 1.** *Rreb1* chimeras display changes in ECM organization.

## Discussion

The transcription factor *Rreb1* is necessary for invertebrate development (*Melani et al., 2008*; *Pickup et al., 2002*; *Wieschaus et al., 1984*; *Wilk et al., 2000*) and is implicated in cancer (*Ferrara and De Vanna, 2016*; *Hui et al., 2019*; *Kent et al., 2017*; *Li et al., 2018*; *Mukhopadhyay et al., 2007*; *Thiagalingam et al., 1996*; *Yao et al., 2019*), suggesting that it plays critical contextual organismal functions. Despite this, we knew little about its role in mammalian development. Here, we demonstrated that *Rreb1* is essential for mouse embryo development. Loss of *Rreb1* resulted in disrupted epithelial architecture of both embryonic and extraembryonic tissues. These defects were consistent with the role of the *Drosophila* homolog of Rreb1, Hindsight (hnt), that regulates cell adhesion during invertebrate development (*Melani et al., 2008*; *Pickup et al., 2002*; *Wilk et al., 2000*). In *Rreb1*$^{-/-}$ mutant embryos and chimeras, pluripotent epiblast cells fell out of their epithelial layer into the space between the epiblast and VE. These events were observed more frequently in chimeras versus *Rreb1*$^{-/-}$ embryos hence, interactions between wild-type and mutant cells, such as differential cell adhesion between these genotypes, may promote invasive-like behaviors. In support of this hypothesis, mathematical models predict that populations with elevated cellular adhesion heterogeneity will exhibit increased tumor cell dissemination (*Reher et al., 2017*). Similarly, loss of hnt in the *Drosophila* retina caused cells to fall out of the epithelium into the under-lying tissue layer (*Pickup et al., 2002*). Thus, Rreb1 is an evolutionarily conserved regulator of tissue architecture.

*Rreb1* homozygous mutant embryos die at midgestation due to a range of cardiovascular defects, including perturbed yolk sac vasculogenesis. These findings are consistent with previous studies that show that Rreb1 ±adult mice have smaller hearts and thickening of the cardiac wall (*Kent et al., 2020*). Moreover, the observed phenotypes are similar to those of VEGF pathway mutants (*Carmeliet et al., 1996*; *Damert et al., 2002*; *Ferrara and De Vanna, 2016*). Loss of *Rreb1* also led to increased expression of H2-Q2, a homolog of HLA-G that regulates VEGF expression through indirect mechanisms (*Liu et al., 2020*) as well as vasculogenesis and differentiation of blood lineages (*Comiskey et al., 2003*; *Liu et al., 2020*). As *Rreb1* functions downstream of, and regulates, receptor tyrosine kinase signaling, its role in vasculogenesis may be mediated via VEGF. Although *Rreb1* was not highly expressed by the yolk sac mesoderm, which will give rise to endothelial cells, it was robustly expressed by the overlying yolk sac endoderm (*Figure 1—figure supplement 1G*). The yolk

sac endoderm secretes factors that regulate cardiogenesis, vasculogenesis, and hematopoiesis (*Arai et al., 1997*; *Belaoussoff et al., 1998*; *Byrd et al., 2002*; *Damert et al., 2002*; *Dyer et al., 2001*; *Goldie et al., 2008*; *Miura and Wilt, 1969*; *Wilt, 1965*). *Rreb1* mutants showed a significant downregulation of genes encoding secreted vasculogenesis-associated factors typically expressed by the VE, as well as genes involved in vesicular transport that form part of the secretory pathway. Thus, the role of *Rreb1* in embryonic vasculogenesis is likely mediated via paracrine interactions with the endoderm.

We previously showed that, in a cancer model, *Rreb1* directly binds to the regulatory region of *Snai1* in cooperation with TGF-β activated SMAD transcription factors to induce the expression of SNAIL, which drives EMT (*Su et al., 2020*). Furthermore, mouse embryos containing *Rreb1*$^{-/-}$ cells exhibit an accumulation of cells at the primitive streak, consistent with a disrupted gastrulation EMT (*Su et al., 2020*). These data suggested that *Rreb1* may be required for EMT in both development and disease contexts. Nevertheless, both *Rreb1*$^{-/-}$ chimeric (*Su et al., 2020*) and mutant embryos did not show a total block to EMT, with cells able to exit the epiblast at the PS and differentiate into the embryonic germ layers. This phenotype is similar to that observed in *Crumbs2* mutant embryos (*Ramkumar et al., 2016*) whereby the initial gastrulation EMT proceeds normally but over time, cells accumulate at the PS. This suggests temporally distinct EMT regulatory mechanisms in vivo with *Crumbs2* and *Rreb1* required for the later stages of this process.

Additionally, upon closer examination we found that loss of *Rreb1* disrupts epithelial architecture. We found that, in the mouse embryo, *Rreb1* is expressed not only in mesenchymal tissues, such as the primitive streak and mesoderm, but also within epithelial tissues such as the trophectoderm, VE and the notochord. Thus, Rreb1 does not drive EMT in all contexts. Likewise, in *Drosophila*, hnt exhibits context-dependent adhesion regulation. For example, loss of hnt in the trachea and retina disrupts epithelial architecture (*Pickup et al., 2002*; *Wilk et al., 2000*), while loss of hnt from border cells results in increased cell-cell adhesion (*Melani et al., 2008*). Thus, its function likely depends on the combination of factors and signaling activities present within any given cell where it is expressed.

Global transcriptional analysis of *Rreb1*$^{-/-}$ embryos revealed that loss of *Rreb1* significantly alters the transcription of cytoskeleton-associated genes, including actin-binding proteins, microtubule components, and microtubule motor proteins. Rreb1 was also shown to directly bind to the loci of a number of cytoskeleton regulators in HEK cell lines (*Kent et al., 2020*). Furthermore, Hnt genetically interacts with and transcriptionally regulates cytoskeleton-associated genes, such as *chickadee* (*Profilin1*), which governs actin polymerization and depolymerization, the F-ACTIN crosslinker *karst* (*Alpha-actinin-1*), Actin-binding protein *jitterbug* (*Filamin A*), a microtubule motor *dynamitin* (*Dynactin2*) and *Rho1*, a GTPase that regulates cytoskeleton organization (*Oliva et al., 2015*; *Wilk et al., 2004*). While the specific factors downstream of *Rreb1* and hnt are distinct, these data suggest a conserved role in cytoskeleton regulation. The transcriptional changes in cytoskeleton regulators corresponded to a change in the organization of the cytoskeleton and adherens junctions whereby wild-type epiblast cell junctions displayed a continuous, linear arrangement of F-ACTIN, E-CADHERIN and Beta-CATENIN, while *Rreb1*$^{-/-}$ exhibited a punctate localization. ACTIN interacts with cadherins (*Han and de Rooij, 2017*) and thus may directly influence their localization. The cytoskeleton mediates vesicular trafficking, which can also regulate E-CADHERIN localization (*Aiello et al., 2018*; *Chen et al., 2003*; *Chung et al., 2014*; *Liang et al., 2015*; *Mary et al., 2002*; *Pilot et al., 2006*; *Sako-Kubota et al., 2014*; *Stehbens et al., 2006*; *Teng et al., 2005*; *Vasileva and Citi, 2018*), and a large number of trafficking genes were upregulated in *Rreb1*$^{-/-}$ embryos. Therefore, a combination of altered vesicle trafficking and/or direct changes in the cytoskeleton may regulate E-CADHERIN localization. As *Rreb1* is not expressed highly within the epiblast, these phenotypes could be due to a loss of low-level epiblast expression or an indirect effect of altered mechanical forces in the embryo stemming from perturbed EMT and, in mutant embryos, the VE. The expression of SNAIL and a number of other EMT and adhesion regulators is mechano-sensitive (*Farge, 2003*; *Pukhlyakova et al., 2018*; *Zhang et al., 2016*), and thus changes in the physical forces within the embryo could underpin ectopic SNAIL expression within a fraction of epiblast cells.

A reduction in ACTIN stress fibers enhances the motility and deformability of cells and is associated with an invasive phenotype in cancer (*Grady et al., 2016*; *Han et al., 2020*; *Katsantonis et al., 1994*; *Suresh, 2007*; *Xu et al., 2012*). Moreover, altered ACTIN organization (*Gloushankova et al., 2017*; *Kovac et al., 2018*) and punctate E-CADHERIN is indicative of an intermediate epithelial-

mesenchymal state, which also correlates with weaker cell-cell adhesion and collective invasion in metastasis (*Aiello et al., 2018*; *George et al., 2017*; *Jolly et al., 2015*; *Saitoh, 2018*). In keeping with this, *Rreb1*$^{-/-}$ cells displayed invasive phenotypes in vivo resulting in ectopic SOX2 +epiblast like cells positioned throughout chimeric embryos. However, ectopic cells were of wild-type and mutant origin indicating that not only cell-autonomous properties, such as cytoskeletal organization, but also cell non-autonomous mechanisms drive this behavior. *Rreb1/hnt* phenotypically interacts with and transcriptionally regulates ECM-associated factors such as *viking* (*Col4a1*), *Cg25c* (*Col4a2*), *Mmp2*, and *Adamts5* (*Deady et al., 2017*; *Wang et al., 2017*; *Wilk et al., 2004*). We also observed a change in the expression of ECM-associated factors in *Rreb1*$^{-/-}$ embryos, some of which have been linked to changes in the metastatic potential of cells. Furthermore, KEGG pathway analysis of down-regulated genes revealed that these were associated with the complement and coagulation cascades, which control a variety of processes, including ECM remodeling, and the corruption of this pathway is linked to cancer metastasis (*Ajona et al., 2019*). Thus, changes in ECM composition in *Rreb1*$^{-/-}$ embryos may drive invasive behaviors. Ectopic SOX2 + cells were associated with abnormal breaks in the basement membrane, elevated levels of Laminin, and Laminin tracks. These ECM tracks are reminiscent of bundles of parallel Collagen fibers, referred to as 'microtracks', observed in cancer. Microtracks are generated through ECM remodeling by invasive leader cells, which subsequently facilitates the migration of less invasive cells within the tumor (*Gaggioli, 2008*; *Gaggioli et al., 2007*; *Poltavets et al., 2018*). Intriguingly, ectopic SOX2 + cells of wild-type origin were adjacent to *Rreb1*$^{-/-}$ cells. Thus, *Rreb1*$^{-/-}$ cells might perform a role comparable to leader cells in cancer metastasis, remodeling the ECM to permit migration of wild-type neighbors. HLA-G (H2-Q2) upregulation is also associated with metastasis and immune cell evasion (*Liu et al., 2020*) and, as a secreted factor, might also promote invasive behaviors in both wild-type and mutant cells.

In sum, we describe phenotypes and cell behaviors in *Rreb1* mutant mouse embryos reminiscent of those observed during cancer cell invasion, including loss of epithelial architecture, aberrant basement membrane breakdown, ECM remodeling, and ectopic exit of cells from an epithelium. The early mouse embryo is an experimentally tractable in vivo system to interrogate these phenotypes and thus, future studies of the function of *Rreb1* in development may also shed light on its role in metastasis and other diseases involving loss of epithelial integrity.

## Materials and methods

### Generation and maintenance of mouse lines

Mice were housed under a 12 hr light-dark cycle in a specific pathogen-free room in the designated facilities of MSKCC. Natural matings were set up in the evening and mice were checked for copulation plugs the following morning. The date of vaginal plug was considered as E0.5. Genotyping was carried out at the time of weaning. Mice were outbred to CD1 animals and maintained on a mixed bred CD-1/129 Sv/C57BL6/C2J background in accordance with the guidelines of the Memorial Sloan Kettering Cancer Center (MSKCC) Institutional Animal Care and Use Committee (IACUC).

To generate the *Rreb1*$^{LacZ}$ reporter mouse line, in vitro fertilization was performed using C57BL/6N-A$^{tm1Brd}$ Rreb1$^{tm1a(EUCOMM)Wtsi}$/WtsiPh (RRID:IMSR_EM:10996) sperm obtained from the European Conditional Mouse Mutagenesis Program (EUCOMM). The Tm1a (knockout-first) allele was genotyped by PCR using the following primers: *Rreb1* 5' arm: CTTCTGTCCCAGAAGCTACATTGC, *Rreb1* 3' arm: GGACAACGGTCACTGAGAAGATGG, Lar3: CAACGGGGTTCTTCTGTTAGTCC and the protocol: Step1–95°C for 3 min, Step 2–35x: 95°C for 30 s, 63°C for 30 s, 72°C for 30 s, Step 3–72°C for 3 min. This results in a wild-type allele amplicon band of 751 bp and a transgenic allele amplicon of 502 bp. Tm1a mice were then crossed with a Flp recombinase mouse line (*Rodríguez et al., 2000*) to remove the neomycin cassette and Exon 6, producing the Tm1b LacZ-tagged null allele. *Rreb1*$^{LacZ/+}$ embryos were analyzed by X-gal staining to determine the *Rreb1* expression pattern.

*Rreb1*$^{-/-}$ mutant mice were generated by CRISPR-mediated genetic knockout. The CRISPR gRNAs used for deleting exon 6 of the *Rreb1* gene were designed using the approach of *Romanienko et al., 2016*. The sequences of the guides are: crRNA#1: TATTATGAACTCCTCTGGAC, crRNA#2: AGTGTCTTCGAAAGAGCCAA, crRNA#3: CGTTACAACAAAGCACCCTT, crRNA#4: AGGAAAACTCGTAGTGGCAC. To initiate cleavage and subsequent deletion of the target locus in mice, guides were injected in pairs, either #1 and #3 or #2 and #4, into the pronuclei of mouse

zygotes at a concentration of 50 ng/μl each, with 100 ng/μl purified Cas9 protein (PNABio, Newbury Park, CA), using conventional techniques (*Behringer et al., 2014*). Founder mice were analyzed for the deletion by PCR using the primers RREB2: GACACCTAGTCACCGAGGAAAC and RREB6: CTG TGGCAGATCTGGTAGGC. This primer pair is located outside of the gRNA cleavage sites, thereby revealing the size of the deletion based on the nucleotide length of the amplicon obtained. The wild-type amplicon size is 1019 bp. The deletion amplicons, if there had been a simple cut and rejoining, would be: Cr#1 and #3: 275 bp. Cr#2 and #4: 456 bp. Genotyping of the *Rreb1* locus was performed by PCR with primers RREB1_1: GTGACAGAGGGAACAGTGGG, RREB1_2: GACACC TAGTCACCGAGGAAAC, RREB1_3: GTGTCTGTGTTGTGCTGCA using the following protocol: Step1–94°C for 3 min, Step 2–35x: 95°C for 30 s, 64°C for 90 s, 72°C for 1 min, Step 3–72°C for 5 min, resulting in a 358 bp amplicon for the wild-type allele and a 275 bp amplicon for the mutant allele. *Rreb1*$^{-/-}$ mice were embryonic lethal at midgestation but no peri-natal lethality was observed for *Rreb1*$^{-/+}$ mice. Therefore, the *Rreb1* mouse line was maintained and *Rreb1*$^{-/-}$ embryos were obtained through heterozygous *Rreb1*$^{-/+}$ intercrosses.

## Generation of chimeric embryos

Approximately 10–15 *Rreb1*$^{-/-}$ ESCs, described in *Su et al., 2020*, harboring a constitutive mCherry fluorescent lineage tracer were injected into E3.5 blastocysts (C57BL/6J, Jackson Laboratory, Bar Harbor, ME) as previously described (*Su et al., 2020*). Injected blastocysts were cultured in KSOM/ AA (Millipore, Billerica, MA) at 37°C in an atmosphere of 5% CO2 to allow for recovery of blastocyst morphology and then implanted into the uterine horns (up to 10 embryos per horn) of E2.5 pseudo-pregnant females (C57BL/6J;CBA F1, Jackson Laboratory) using standard protocols. Chimeric embryos were recovered between E7.5-E9.5.

## Wholemount in situ hybridization

To produce the *Rreb1* riboprobes, RNA was isolated from pooled E12.5 CD1 mouse embryos using an RNeasy Plus Mini Kit (Qiagen, Hilden, Germany) and then used to generate cDNA with a Quanti-Tect Reverse Transcription Kit (Qiagen), as per manufacturer's instructions. Primers (5' UTR L: GGGCCTTTGTCTCATGCTCC, 5' UTR R: CGCAGAATGTTTTCCTCAACAG) were designed against a unique 502 bp region within the *Rreb1* 5' UTR and used to PCR amplify this fragment from E12.5 embryo cDNA. The PCR product was purified using a QIAquick PCR Purification Kit (Qiagen) and a TOPOTA Cloning Kit (K461020, Thermo Fisher Scientific) used to introduce the fragment into a pCRII-TOPO Vector and transformed into *E. coli*. Colonies were picked, expanded and the plasmid isolated for sequencing. A plasmid containing the correct sequence (5'-CGCAGAATGTTTTCCTCAA-CAGTTGACAATTTTAGGATAAATAGAACTTTAGAAAAATTACTACTATCAATCATCTAAGTA TTCCGAATAGGAAAAAAAGTCAAAATAAGTAAGGGACGCTGGAGCTACCTCAGTGAAGGG-GAAAAAATATCCAATCCCACTTTTCTGTATTACATGTGTGGTAGCTAAAGAACTCCATAGAATG TTCAAAAAAAAAAAAAAAAGACGGCACTGAAGATTATCATGTCAAAGCACCAAGCTCATTACA TCACTGTTACCTTAATGCAAAGTCCCACTTCTCCGGAATGGCCTCCATACTTAGAAACTC TTGGAACTTGTCAGGCAAAGGTTATGGGGAGGGAAGTGAAGGAGCCTATGACCACTGTCACTG TGTCTGATACATTTATTTACAGATAAGCCTTGGTGGCTCAGACCCACAGGCACAGATTATA TGGAAAGTAACAGCCTGTGACTTCTGAGACAAAGAATGGAGCATGAGACAA-3') was selected, lin-earized and the dual promoter system within the pCRII-TOPO Vector used to amplify and DIG label both a control sense and an antisense probe. Wholemount mRNA in situ hybridization was then carried out as previously reported (*Conlon and Rossant, 1992*).

## X-gal staining

X-gal staining of cells and embryos containing the *Rreb1*-LacZ reporter was performed using a β-Gal Staining Kit (K146501, Invitrogen, Waltham, MA) as per manufacturer's instructions. Embryos and cells were fixed for 15 mins at room temperature followed by staining until the blue color was detectable (2–3 hr) at 37°C.

## Cell lines and cell culture

Cells were maintained in standard serum/LIF ESC medium (Dulbecco's modified Eagle's medium (DMEM) (Gibco, Gaithersburg, MD) containing 0.1 mM non-essential amino-acids (NEAA), 2 mM

glutamine and 1 mM sodium pyruvate, 100 U/ml Penicillin, 100 µg/ml Streptomycin (all from Life Technologies, Carlsbad, CA), 0.1 mM 2-mercaptoethanol (Sigma, St. Louis, MO), and 10% Fetal Calf Serum (FCS, F2442, Sigma) and 1000 U/ml LIF) as previously described (*Morgani et al., 2018a*). C57BL/6N-A$^{tm1Brd}$ Rreb1$^{tm1a(EUCOMM)Wtsi}$/WtsiPh (RRID:IMSR_EM:10996) embryonic stem (ES) cell lines were used to analyze *Rreb1* expression and also converted to an epiblast stem cell (EpiSC) state through prolonged culture (more than five passages) in N2B27 medium containing 12 ng/ml FGF2 (233-FB-025, R and D Systems) and 20 ng/ml ACTIVIN A (120–14P, Peprotech, Rocky Hills, NJ), as previously described (*Tesar et al., 2007*). ES cells were routinely tested and found to be mycoplasma negative. They were validated by genotyping and used to generate germline transmitting mouse chimeras.

## Immunostaining

Cell lines were immunostained as previously described (*Morgani et al., 2018a*). Post-implantation embryos were fixed in 4% paraformaldehyde (PFA) for 15 min at room temperature (RT). Embryos were washed in phosphate-buffered saline (PBS) plus 0.1% Triton-X (PBST-T) followed by 30 min permeabilization in PBS with 0.5% Triton-X. Embryos were washed in PBS-T and then blocked overnight at 4°C in PBS-T, 1% bovine serum albumin (BSA, Sigma) and 5% donkey serum. The following day, embryos were transferred to the primary antibody solution (PBS-T with appropriate concentration of antibody) and incubated overnight at 4°C. The next day, embryos were washed 3 × 10 min in PBS-T and then transferred to blocking solution at RT for a minimum of 5 hr. Embryos were transferred to secondary antibody solution (PBS-T with 1:500 dilution of appropriate secondary conjugated antibody) and incubated overnight at 4°C. Embryos were then washed 3 × 10 min in PBS-T with the final wash containing 5 µg/ml Hoechst. Where F-ACTIN staining was performed, Alexa Fluor conjugated phalloidin (Thermo Fisher Scientific, Waltham, MA) was added to the primary and secondary antibody solutions at a 1:500 dilution.

## Antibodies

The following primary antibodies were used in this study: Beta-catenin (RRID:AB_397555, BD Transduction labs, Billerica, MA, 610154, 1:500), Brachyury (RRID:AB_2200235, R and D, AF2085, 1:100), CD31 (RRID:AB_394819, BD Biosciences, 553373, 1:100) CD105 (RRID:AB_354735, R and D Systems, AF1320, 1:100), E-cadherin (RRID:AB_477600, Millipore Sigma, U3254, 1:200), Endoglin (CD105) (RRID:AB_354735, R and D Systems, AF1320, 1:300), Flk-1 (RRID:AB_355500, R and D Systems, AF644, 1:200) Gata6 (RRID:AB_10705521, D61E4 XP, Cell Signaling, 5851, 1:500), GFP (RRID:AB_300798, Abcam, ab13970), Laminin (RRID:AB_477163, Millipore Sigma, L9393, 1:500), N-cadherin (RRID:AB_2077527, BD Biosciences, 610920, 1:200), Pecam-1 (RRID:AB_394819, BD Biosciences, 553373, 1:200), RFP (Rockland, Limerick, PA, 600-400-379, 1:300), Snail (RRID:AB_2191738, R and D Systems, AF3639, 1:50), Sox2 (RRID:AB_11219471, Thermo Fisher Scientific, 14-9811-82, 1:200), Sox17 (RRID:AB_355060, R and D Systems, AF1924, 1:100), ZO-1 (RRID:AB_87181, Invitrogen, 33–9100, 1:200).

## Cryosectioning

Embryos were oriented as desired and embedded in Tissue-Tek OCT (Sakura Finetek, Japan). Samples were frozen on dry ice for approximately 30 min and subsequently maintained for short periods at −80°C followed by cryosectioning using a Leica CM3050S cryostat. Cryosections of 10 µm thickness were cut using a Leica CM3050S cryostat and mounted on Colorfrost Plus microscope slides (Fisher Scientific) using Fluoromount G (RRID:SCR_015961, Southern Biotech, Birmingham, AL) and imaged using a confocal microscope as described.

## Confocal imaging and quantitative image analysis

Embryos were imaged on a Zeiss LSM880 laser scanning confocal microscope. Whole-mount embryos were imaged in glass-bottom dishes (MatTek, Ashland, MA) in PBS. Raw data were processed in ImageJ open-source image processing software (Version: 2.0.0-rc-49/1.51d).

Nuclei orientation (*Figure 5—figure supplement 1E–G*) was measured manually using Fiji (RRID:SCR_002285, Image J) software. Using the angle tool, we measured the angle between the long axis of individual epiblast nuclei and the underlying basement membrane, marked by Laminin staining on

confocal optical sections of transverse cryosections. We measured the angle of 143 cells from 3 *Rreb1*$^{+/+}$ embryos and 136 cells from 3 *Rreb1*$^{-/-}$ embryos.

We quantified proliferation in *Rreb1*$^{+/+}$ versus *Rreb1*$^{-/-}$ embryos (*Figure 5—figure supplement 1L*) by manually counting the number of phosphorylated histone H3 (pHH3) positive cells in the epiblast, outer endoderm layer or wings of mesoderm in transverse cryosections of *Rreb1*$^{+/+}$ or *Rreb1*$^{-/-}$ embryos. Initially, cell counts were also categorized as divisions in anterior versus posterior embryonic regions but, as no differences were observed, these data were subsequently combined. We performed counts on cryosections comprising three entire embryos per genotype. Data was analyzed as the absolute numbers of dividing cells per cell type. Additionally, we counted the total number of cells per cell type per section and normalized the number of dividing cells to this value to account for differences based on embryo or tissue size. Statistics were performed on a per embryo rather than a per cell basis.

The level of GFP in the VE of *Afp*-GFP; *Rreb1*$^{+/+}$ and *Rreb1*$^{-/-}$ embryos was quantified by manually selecting the embryonic and extraembryonic region of confocal maximum intensity projection images and measuring the mean fluorescence intensity using Fiji software.

Quantification of SOX2 protein levels (*Figure 6—figure supplement 1F*) were carried out on cryosections of *Rreb1*$^{-/-}$ chimeric embryos containing cells expressing high levels of SOX2 (SOX2$^{HI}$ cells) to determine the approximate fold change in protein level relative to normal surrounding cells. To make measurements, nuclei were manually identified using the freehand selection tool in Fiji software. Aberrant SOX2$^{HI}$ cells could readily be distinguished from standard neighboring cells by their elevated signal after immunostaining for SOX2 protein. Mean fluorescence intensity of SOX2 immunostaining was measured within all SOX2$^{HI}$ nuclei within a particular cryosection and an equivalent number of randomly selected nuclei with normal SOX2 expression within the anterior and posterior epiblast regions were measured. Mean SOX2 fluorescence intensity in each nucleus was normalized to the corresponding mean fluorescence intensity of the Hoechst nuclear stain. All data is shown relative to the mean SOX2 fluorescence intensity measured in 'normal' anterior epiblast cells of the same confocal optical section. A total of 8 embryos, 35 cryosections, and 696 cells were analyzed. Statistics were carried out on the average fluorescence levels per embryo.

The localization of SOX2$^{HI}$ cells (identified manually from SOX2 immunostaining) (*Figure 6—figure supplement 1G*) was scored based on their location within confocal images of cryosectioned *Rreb1*$^{-/-}$ chimeric embryos. Scoring was carried out on 76 cryosections from seven independent embryos that contained high numbers of SOX2$^{HI}$ cells. SOX2$^{HI}$ cells were scored as being within the Epi itself, at the Epi-VE interface (outside of the epiblast epithelium), within the primitive streak or wings of mesoderm (mesoderm) or within the amniotic cavity.

## Statistics

Statistical analysis of significance was assessed using a one-way ANOVA (p<0.0001) followed by unpaired *t*-tests to compare particular groups (GraphPad Prism, RRID:SCR_002798, GraphPad Software, Inc, Version 7.0a).

## RNA-sequencing and data analysis

Frozen tissue was homogenized in TRIzol Reagent (ThermoFisher catalog # 15596018) using the QIAGEN TissueLyser at 15 Hz for 2–3 min with a Stainless-Steel Bead (QIAGEN catalog # 69989). Phase separation was induced with chloroform. RNA was precipitated with isopropanol and linear acrylamide and washed with 75% ethanol. The samples were resuspended in RNase-free water. After RiboGreen quantification and quality control by Agilent BioAnalyzer, 150 g of total RNA underwent polyA selection and TruSeq library preparation according to instructions provided by Illumina (TruSeq Stranded mRNA LT Kit, catalog # RS-122–2102), with 8 cycles of PCR. Samples were barcoded and run on a HiSeq 4000 in a 50 bp/50 bp paired-end run, using the HiSeq 3000/4000 SBS Kit (Illumina). An average of 47 million paired reads was generated per sample. The percent of mRNA bases averaged 67%.

The output data (FASTQ files) were mapped to the target genome using the rnaStar aligner (*Dobin et al., 2013*) that maps reads genomically and resolves reads across splice junctions. We used the two pass mapping method outlined in *Engström et al., 2013*, in which the reads are mapped twice. The first mapping pass uses a list of known annotated junctions from Ensemble.

Novel junctions found in the first pass were then added to the known junctions and a second mapping pass is done (on the second pass the RemoveNoncanoncial flag is used). After mapping, we post-processed the output SAM files using the PICARD tools to: add read groups, AddOrReplaceReadGroups which in additional sorts the file and converts it to the compressed BAM format. We then computed the expression count matrix from the mapped reads using HTSeq (https://www.huber.embl.de/users/anders/HTSeq/doc/overview.html) and one of several possible gene model databases. The raw count matrix generated by HTSeq was then processed using the R/Bioconductor package DESeq (https://www.huber.embl.de/users/anders/DESeq/) which is used to both normalize the full dataset and analyze differential expression between sample groups. The data was clustered in several ways using the normalized counts of all genes that a total of 10 counts when summed across all samples; 1. Hierarchical cluster with the correlation metric (Dij = 1 - cor(Xi,Xj)) with the Pearson correlation on the normalized log2 expression values. 2. Multidimensional scaling. 3. Principal component analysis. Heatmaps were generated using the heatmap.2 function from the gplots R package. For the Heatmaps the top *100* differentially expressed genes are used. The data plot represents the mean-centered normalized log2 expression of the top 100 significant genes. We ran a gene set analysis using the GSA package with gene sets from the Broads mSigDb. The sets used were: Mouse: c1, c2, c3, c4, c5. Gene ontology analyses were performed using the Database for Annotation, Visualization, and Integrated Discovery (DAVID) Bioinformatics resource (Version 6.8) gene ontology functional annotation tool (http://david.abcc.ncifcrf.gov/tools.jsp) with all NCBI *Mus musculus* genes as a reference list. KEGG pathway analysis was performed using the KEGG Mapper – Search Pathway function (https://www.genome.jp/kegg/tool/map_pathway2.html). We performed a manual literature search to determine the proportion of significantly changing genes associated with cancer progression and metastasis.

## Accession numbers

The Gene Expression Omnibus accession number for the RNA-sequencing data reported in this study is GSE148514.

## Acknowledgements

We thank members of the Hadjantonakis and Massagué labs for critical discussions and comments on the manuscript. We also thank members of MSKCC's Mouse Genetics and Integrated Genomics Operation (IGO) and Bioinformatics Core facilities. All cores are funded by the NCI Cancer Center Support Grant (CCSG, P30 CA08748) and IGO is additionally funded by Cycle for Survival, and the Marie-Josée and Henry R Kravis Center for Molecular Oncology. SMM was supported by a Wellcome Trust Sir Henry Wellcome postdoctoral fellowship under the supervision of JN and AKH. Work in the Hadjantonakis lab was supported by grants from the NIH (R01HD086478, R01HD094868 and R01DK084391), work in the Massagué lab was supported by grants from the NIH (R01CA34610 and R35CA252878), and both labs were supported by NIH P30CA008748. J Massagué is a stockholder in Sholar Rock, Inc, a TGF-beta company whose work is unrelated to the present work.

## Additional information

### Funding

| Funder | Grant reference number | Author |
|---|---|---|
| Wellcome Trust | 110151/Z/15/Z | Sophie M Morgani |
| National Institutes of Health | R01HD086478 | Anna-Katerina Hadjantonakis |
| National Institutes of Health | R01HD094868 | Anna-Katerina Hadjantonakis |
| National Institutes of Health | R01DK084391 | Anna-Katerina Hadjantonakis |
| National Institutes of Health | R01CA34610 | Joan Massagué |
| National Institutes of Health | R35CA252878 | Joan Massagué |
| National Cancer Institute | P30 CA08748 | Joan Massagué Anna-Katerina Hadjantonakis |

The funders had no role in study design, data collection and interpretation, or the decision to submit the work for publication.

### Author contributions
Sophie M Morgani, Conceptualization, Data curation, Formal analysis, Funding acquisition, Validation, Investigation, Visualization, Methodology, Writing - original draft, Writing - review and editing; Jie Su, Conceptualization, Investigation, Methodology, Writing - review and editing; Jennifer Nichols, Resources, Supervision, Writing - review and editing; Joan Massagué, Conceptualization, Resources, Supervision, Writing - review and editing; Anna-Katerina Hadjantonakis, Conceptualization, Resources, Supervision, Funding acquisition, Writing - review and editing

### Author ORCIDs
Sophie M Morgani (iD) https://orcid.org/0000-0002-4290-1080
Jie Su (iD) https://orcid.org/0000-0002-3071-7139
Jennifer Nichols (iD) https://orcid.org/0000-0002-8650-1388
Joan Massagué (iD) https://orcid.org/0000-0001-9324-8408
Anna-Katerina Hadjantonakis (iD) https://orcid.org/0000-0002-7580-5124

### Ethics
Animal experimentation: Animal experimentation: Animal experimentation: All mice used in this study were maintained in accordance with the guidelines of the Memorial Sloan Kettering Cancer Center (MSKCC) Institutional Animal Care and Use Committee (IACUC) under protocol number 03-12-017 (PI Hadjantonakis).

### Decision letter and Author response
Decision letter https://doi.org/10.7554/eLife.64811.sa1
Author response https://doi.org/10.7554/eLife.64811.sa2

## Additional files
### Supplementary files
• Transparent reporting form

### Data availability
Sequencing data have been deposited in GEO under accession codes GSE148514. Source data files for Figure 3 have been provided.

The following dataset was generated:

| Author(s) | Year | Dataset title | Dataset URL | Database and Identifier |
|---|---|---|---|---|
| Morgani SM, Su J, Nichols J, Massagué J, Hadjantonakis A-K | 2020 | RNA-sequencing of Rreb1+/+ and Rreb1-/- embryonic day 7.5 mouse embryos | http://www.ncbi.nlm.nih.gov/geo/query/acc.cgi?acc=GSE148514 | NCBI Gene Expression Omnibus, GSE148514 |

The following previously published datasets were used:

| Author(s) | Year | Dataset title | Dataset URL | Database and Identifier |
|---|---|---|---|---|
| Pijuan-Sala B, Griffiths JA, Guibentif C, Hiscock TW, Jawaid W, Calero-Nieto FJ, Mulas C, Ibarra-Soria X, Tyser RCV, | 2019 | Timecourse single-cell RNAseq of whole mouse embryos harvested between days 6.5 and 8.5 of development | https://www.ebi.ac.uk/arrayexpress/experiments/E-MTAB-6967/ | ArrayExpress, E-MTAB-6967 |

| | | | | |
|---|---|---|---|---|
| Ho DLL, Reik W, Srinivas S, Simons BD, Nichols J, Marioni JC, Göttgens B | | | | |
| Pijuan-Sala B, Griffiths JA, Guibentif C, Hiscock TW, Jawaid W, Calero-Nieto FJ, Mulas C, Ibarra-Soria X, Tyser RCV, Ho DLL, Reik W, Srinivas S, Simons BD, Nichols J, Marioni JC, Göttgens B | 2019 | Single-cell RNA-seq of mouse endothelial cells from three distinct embryonic locations | https://www.ebi.ac.uk/arrayexpress/experiments/E-MTAB-6970/ | ArrayExpress, E-MTAB-6970 |
| Pijuan-Sala B, Griffiths JA, Guibentif C, Hiscock TW, Jawaid W, Calero-Nieto FJ, Mulas C, Ibarra-Soria X, Tyser RCV, Ho DLL, Reik W, Srinivas S, Simons BD, Nichols J, Marioni JC, Göttgens B | 2019 | Single-cell RNAseq of control chimeric mouse embryos at embryonic days 7.5 and 8.5 of mouse development | https://www.ebi.ac.uk/arrayexpress/experiments/E-MTAB-7324/ | ArrayExpress, E-MTAB-7324 |
| Pijuan-Sala B, Griffiths JA, Guibentif C, Hiscock TW, Jawaid W, Calero-Nieto FJ, Mulas C, Ibarra-Soria X, Tyser RCV, Ho DLL, Reik W, Srinivas S, Simons BD, Nichols J, Marioni JC, Göttgens B | 2019 | Single-cell RNAseq of Tal1 knockout chimeric mouse embryos at embryonic day 8.5 of mouse development | https://www.ebi.ac.uk/arrayexpress/experiments/E-MTAB-7325/ | ArrayExpress, E-MTAB-7325 |
| Nowotschin S, Setty M, Kuo Y-Y, Liu V, Garg V, Sharma R, Simon CS, Saiz N, Gardner R, Boutet SC, Church DM, Hoodless PA, Hadjantonakis A-K, Pe'er D | 2019 | The emergent landscape of the mouse gut endoderm at single-cell resolution | https://www.ncbi.nlm.nih.gov/geo/query/acc.cgi?acc=GSE123046 | NCBI Gene Expression Omnibus, GSE123046 |
| Nowotschin S, Setty M, Kuo Y-Y, Liu V, Garg V, Sharma R, Simon CS, Saiz N, Gardner R, Boutet SC, Church DM, Hoodless PA, Hadjantonakis A-K, Pe'er D | 2019 | The emergent landscape of the mouse gut endoderm at single-cell resolution | https://www.ncbi.nlm.nih.gov/geo/query/acc.cgi?acc=GSE123124 | NCBI Gene Expression Omnibus, GSE123124 |

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
