## [Decision Letter]

**Acceptance summary:**

This manuscript reports on the phenotypic analyses of mice mutant for the Rreb1 gene encoding transcription factor implicated in cancer. Extending their earlier report that Rreb1 is essential for mouse embryonic development and normal epithelial mesenchymal transition during gastrulation, this manuscript documents a broad expression and contextual functions of Rreb1 in embryonic and extraembryonic tissues. Evidence is presented that Rreb1 affects many developmental processes, including gastrulation cell movements and tissue architecture, and does not drive epithelial mesenchymal transition in all these contexts.

**Decision letter after peer review:**

Thank you for submitting your article "The transcription factor Rreb1 regulates epithelial architecture and invasiveness in gastrulating mouse embryos" for consideration by *eLife*. Your article has been reviewed by 2 peer reviewers, one of whom is a member of our Board of Reviewing Editors, and the evaluation has been overseen by Kathryn Cheah as the Senior Editor. The reviewers have opted to remain anonymous.

Summary:

This manuscript finds that Rreb1, a Ras-responsive transcription factor often mutated in cancer, is broadly expressed in pre- and post-implantation mouse embryo and essential for proper development of multiple tissues and structures. Through analyses of Rreb1 mouse mutants the authors provide a detailed characterization of the effects of Rreb1 loss on epithelial integrity during gastrulation, including ectopic exit of cells from the epithelial epiblast and remodeling of the extracellular matrix reminiscent of cancer metastasis. Although the precise relationship between Rreb1 and epithelial-mesenchymal transition and cell fates specification in different regions of the gastrula epiblast remain to be determined, Rreb1 does not drive epithelial mesenchymal transition in all these contexts. The implications of these findings for metastasis make this manuscript of interest to both developmental and cancer biologists.

Essential Revisions:

This manuscript reports on the phenotypes observed in mouse Rreb1 mutants. Rreb1 is a transcription factor known for its role in cancer and cancer metastasis. In their previous work studies of chimeric embryos containing Rreb1-/- ES cells, the authors demonstrated that Rreb1 is essential for mouse embryonic development: Rreb1 mutant epiblast presents with defects typical of gastrulation failure with mutant cells in the primitive streak showing impaired but not completely blocked epithelial mesenchymal transition (EMT) (Su et al., Nature, 2020). In the current manuscript a broad Rreb1 expression at pre- and post-implantation stages is documented. In addition to gastrulation anomalies, Rreb1 mouse mutants also manifest defects in organogenesis, including defective notochord formation, neurulation and cardiovascular development. RNA-seq studies revealed upregulation of many cytoskeletal and extracellular matrix components. Based on histological and immunohistological analyses, it is proposed that germ layer formation proceeds on schedule but some pluripotent Rreb1-/- epiblast cells acquired mesenchymal characteristics, including ectopic expression of SNAIL, precocious exit from the epiblast epithelium via basement membrane, which appears broken in ectopic positions. Nice chimera experiments suggest that these defects are due cell-autonomous changes in the cytoskeleton and non-cell-autonomous changes in the ECM. The overall conclusion is that Rreb1 has pleiotropic and diverse functions in morphogenesis of the embryonic and extraembryonic tissues, and that Rreb1 does not drive EMT in all contexts. That Rreb1 can have different roles in different cellular context has been demonstrated in *Drosophila*. One wonders whether these modest conceptual advances in understanding of Rreb1 function during mouse embryonic development warrant publication in *eLife* or manuscript is suitable in a more specialized journal. There are also questions about the interpretation of the Rreb1 mutant defects as affecting architectural and morphogenetic properties of epiblast cells during gastrulation without altering their fates.

1. One is concerned that the Rreb1 mutant primitive streak analyses reported in the current manuscript appear contrary to the EMT defects postulated by Su et al., Nature, 2020. In the current manuscript, it is stated "… Rreb1-/- epiblast cells underwent an EMT at the primitive streak, delaminated from the epithelium, and migrated anteriorly in the wings of mesoderm (Figure 5E)." Moreover, Su et al., 2020 strongly indicated a pro-EMT function for Rreb1 in embryoid bodies. Particularly, transcriptomic profiling of Rreb1-/- gastrulae does not appear to overlap well with Rreb1-dependent genes identified in embryoid bodies, especially markers of EMT and mesoderm. It will be important for the authors to acknowledge/address these differences between models in their discussion of Rreb1 function during gastrulation.

2. To illuminate the proposed different roles of Rreb1 during mouse gastrulation, a more precise characterization of the cellular phenotypes and molecular defects in the Rreb1 mutants in different region of the epiblast would be required. What are the identities of the epiblast cells that exhibit ectopic mesenchymal morphology, expression of SNAIL and precious exit from the epithelium? Some of such abnormally behaving cells expressed the epiblast marker *SOX2*, what makes the authors conclude that they continue to exhibit a more pluripotent epiblast fate and have not acquired mesodermal or endodermal fates. However, this should be investigated with additional mesendodermal markers and Nodal activity reporters. This is important, as the current focus of the manuscript is on the effect Rreb1 LOF has on expression of cytoskeletal and ECM components. Bulk RNA seq experiment are unlikely to reveal subtle changes in expression of SNAIL, or other mesendodermal markers that are expressed in primitive streak. A more thorough characterization of *Sox2* Hi cells and "delaminating" cells will also be important, as will clarify apparent differences between Rreb1 mutant and chimeric embryos. Do the observed morphogenetic phenotypes occur without any changes in cell fate specification?

3. No specific molecular link is demonstrated between Rreb1 and any of the reported phenotypes. However, in the Discussion the authors refer to their previous work "We previously showed that, in a cancer model, Rreb1 directly binds to the regulatory region of Snai1 in cooperation with TGF-β activated SMAD transcription factors to induce the expression of SNAIL, which drives EMT (Su et al., 2020)." Yet, in the mouse Rreb1 mutant epiblast, ectopic expression of SNAIL is observed. As Rreb1 can act as transcriptional activator and repressor, such distinct activity in the presence or absence of TGFb-signaling is plausible and experiments testing such models would provide mechanistic insights into these varied functions of Rreb1 during mouse gastrulation. This also underscores the need to more carefully evaluate Nodal signaling and cell fate specification in Rreb1 mutants.

4. The precocious delamination of epiblast cells from epithelia is associated with abnormal ECM organization and broken down basement membrane in ectopic locations in addition to the primitive streak. Kyprianou et al., Nature, 2020 showed that the basement membrane remodeling in the primitive streak region is associated with expression of *MMP2* and MMP14 downstream of Nodal signaling. Is expression of these enzymes affected in Rreb1 mutant gastrulae?

5. As discussed above, Rreb1 mutant embryos and Rreb1/WT chimeras appear to exhibit distinct phenotypes. Can the authors explain or speculate about these phenotypic differences between the two conditions? The authors should determine/report whether breaches of the basement membrane and ECM tracks are observed in Rreb1 mutants similar to chimeric embryos. One would expect to see these near the chains of cells that cross boundaries between germ layers, for example.

5. Interesting vascular phenotypes have been described in the yolk sac, but their cellular and molecular mechanisms are not clear. It would be important to understand the common and unique cellular endpoints that are regulated by this interesting protein. This should be at least discussed.

6. Based on Figure 1B in which the authors investigate Rreb1 expression in pluripotent ES and epiblast stem cells, it is concluded that "Rreb1 is expressed in the embryonic lineages as pluripotency is exited and the germ layers are specified". Can this be concluded given the strong Rreb1 expression seen in the ICM of pre-implantation stage embryos? More nuanced conclusion should be considered.

7. Rreb1 was shown to promote Snai1 expression in Su et al., yet here Snai1 is ectopically expressed upon loss of Rreb1. The authors speculate that changes in epithelial architecture within the VE of Rreb1 mutants leads to non-specific antibody staining (as with N-cadherin, Figure 5). Is it possible that the very bright Snai1 staining in isolated cells leaving the epiblast (Figure 6) is similarly non-specific dues to localized disruption of epithelial architecture? Is Snai1 staining observed at sites of basement membrane breach in chimeric embryos?

8. In the discussion (page 14, lines 25-27), the authors speculate that Rreb1 expression within the VE may be important for epithelial integrity in the epiblast… but isn't this inconsistent with results from chimeric embryos, in which WT VE does not prevent ectopic delamination of mutant (and WT) epiblast cells?

9. For several phenotypes, including notochord defects (Figure S2), punctate localization of F-actin and E-cadherin (Figure 4), and Afp-GFP localization (Figure 3), only 1 example is shown and no sample numbers are reported, making it difficult to evaluate how representative these images are of mutant phenotypes. The authors should include the number of embryos examined for every experiment shown.

---

## [Author Response]

Essential Revisions:This manuscript reports on the phenotypes observed in mouse Rreb1 mutants. Rreb1 is a transcription factor known for its role in cancer and cancer metastasis. In their previous work studies of chimeric embryos containing Rreb1-/- ES cells, the authors demonstrated that Rreb1 is essential for mouse embryonic development: Rreb1 mutant epiblast presents with defects typical of gastrulation failure with mutant cells in the primitive streak showing impaired but not completely blocked epithelial mesenchymal transition (EMT) (Su et al., Nature, 2020). In the current manuscript a broad Rreb1 expression at pre- and post-implantation stages is documented. In addition to gastrulation anomalies, Rreb1 mouse mutants also manifest defects in organogenesis, including defective notochord formation, neurulation and cardiovascular development. RNA-seq studies revealed upregulation of many cytoskeletal and extracellular matrix components. Based on histological and immunohistological analyses, it is proposed that germ layer formation proceeds on schedule but some pluripotent Rreb1-/- epiblast cells acquired mesenchymal characteristics, including ectopic expression of SNAIL, precocious exit from the epiblast epithelium via basement membrane, which appears broken in ectopic positions. Nice chimera experiments suggest that these defects are due cell-autonomous changes in the cytoskeleton and non-cell-autonomous changes in the ECM. The overall conclusion is that Rreb1 has pleiotropic and diverse functions in morphogenesis of the embryonic and extraembryonic tissues, and that Rreb1 does not drive EMT in all contexts. That Rreb1 can have different roles in different cellular context has been demonstrated in *Drosophila*. One wonders whether these modest conceptual advances in understanding of Rreb1 function during mouse embryonic development warrant publication in eLife or manuscript is suitable in a more specialized journal. There are also questions about the interpretation of the Rreb1 mutant defects as affecting architectural and morphogenetic properties of epiblast cells during gastrulation without altering their fates.

We would like to thank the reviewers for their thorough analysis of our manuscript. While the role of the *Drosophila* homolog of Rreb1 (Hnt) has been previously studied, little was known about the role of Rreb1 in a mammalian, non-disease state, and nothing has been previously reported about its expression and function in mammalian development. Thus, the findings reported in this study are novel and provide the starting point for future investigations into the panoply of Rreb1 contextual functions. Furthermore, our observation that loss of Rreb1 results in a change in the localization of e-cadherin protein at adherens junctions has not been previously reported in any model system in which Rreb1 or a homolog has been studied, and importantly these findings have implications for understanding the contribution of this factor to disease states. As Rreb1 has been linked to various human cancers, characterizing its role in mammalian contexts is of critical importance. Therefore, our observation that Rreb1 is essential for mouse embryonic development, where it maintains epithelial architecture, is an important insight, potentially related to its role in cancer. As such, we believe that this study is of interest to the broad readership of *eLife*.

1. One is concerned that the Rreb1 mutant primitive streak analyses reported in the current manuscript appear contrary to the EMT defects postulated by Su et al., Nature, 2020. In the current manuscript, it is stated "… Rreb1-/- epiblast cells underwent an EMT at the primitive streak, delaminated from the epithelium, and migrated anteriorly in the wings of mesoderm (Figure 5E)." Moreover, Su et al., 2020 strongly indicated a pro-EMT function for Rreb1 in embryoid bodies. Particularly, transcriptomic profiling of Rreb1-/- gastrulae does not appear to overlap well with Rreb1-dependent genes identified in embryoid bodies, especially markers of EMT and mesoderm. It will be important for the authors to acknowledge/address these differences between models in their discussion of Rreb1 function during gastrulation.

We apologize for any confusion, the phenotypes of Rreb1^-/-^ chimeric embryos we described previously in Su et al., Nature 2020, and the phenotypes of Rreb1^-/-^ embryos described in this study are comparable. Based on the reviewer’s comment, we acknowledge that this may not have been clear and, as such, have altered the text and added additional data to clarify this important point. We hope that this addresses the reviewer’s concerns.

In Su et al., Nature, 2020, we analyzed embryonic day (E) 7.5 and 8.5 chimeric embryos containing Rreb1^-/-^ ESC-derived epiblast cells and reported that:

“While mutant embryo chimeras expressed T/Brachyury and SNAIL within the PS and nascent mesoderm (in both WT and Rreb1^−/−^ cells), they frequently showed an accumulation of cells in the posterior epiblast resulting in bulges into the amniotic cavity and/or a folded epiblast layer containing multiple cavities (Figure 4h–i, Extended Data Figure 10f–g)…… Notably, Rreb1 mutant cells did not exhibit an absolute EMT block.”

Thus, in both gastrula stage chimeras and mutant embryos, Rreb1^-/-^ cells express primitive streak markers, undergo EMT, and undergo an initial migration anteriorly within the wings of mesoderm (Su et al., Figure 4j). As also observed in the present study in Rreb1 mutant embryos, we showed that chimeric embryos containing Rreb1^-/-^ cells exhibit abnormal folding of the epiblast (Figure 4h and i and Extended Data Figure 10f and g), and ectopic expression of the EMT regulator SNAIL, resulting in what appeared to be multiple primitive streak-like regions and partial axis duplications (Su et al., Figure 4g, i, Extended Data Figure 10e, f). Thus, chimeric embryos containing Rreb1^-/-^ cells (Su et al., 2020) and Rreb1^-/-^ embryos display comparable phenotypes. To avoid confusion, we have modified the manuscript text to describe the previous phenotype in more detail (P5):

“Previously, we generated chimeric embryos by injecting Rreb1^-/-^ ESCs into wild-type host embryos. While Rreb1^-/-^ cells could undergo the gastrulation EMT, initially migrate within the wings of mesoderm, and initiate differentiation into germ layer derivatives, cells accumulated at the primitive streak over time suggesting that the gastrulation EMT is a continuum of distinct phases and that later EMT events are perturbed (Su et al., 2020)”

Furthermore, while in vitro stem cell models represent valuable, tractable, and simplified tools to study early developmental events, the in vivo reality is far more complex emphasizing the importance of comparing in vitro observations to in vivo. As the reviewer’s discuss, in Su et al., 2020, in vitro embryoid body experiments suggest that Rreb1 is necessary for EMT in embryonic stem cell (ESC) systems. However, in vivo we observed that EMT was clearly perturbed (at least its initiation) but not entirely blocked. The partial EMT block observed in embryos containing Rreb1 mutant cells is similar to that observed by others in Crumbs2 mutant embryos (also reported as having gastrulation EMT defects), whereby the first stages of gastrulation EMT proceed normally and hence the first round of cells can exit the PS and differentiate, but EMT defects emerge and accumulate as gastrulation proceeds (around E7.5) (Ramkumar et al., 2016). These data suggest that there are distinct, stage-specific mechanisms regulating EMT during the progression of gastrulation and that Crumbs2 and Rreb1 likely play a role in the later EMT, or that there is greater redundancy in factors directing the regulation of the initiation of a gastrulation EMT in mice compared to ESC systems. It is currently unknown how these distinct EMT mechanisms in vivo relate to those employed in vitro during ESC differentiation, and whether ESC differentiation captures the full spectrum of the gastrulation EMT. However, based on our in vivo phenotypes and the defects observed in Rreb1^-/-^ embryoid body differentiation in vitro (Su et al., 2020), one could hypothesize that this in vitro EMT is regulated by mechanisms similar to the later in vivo EMT events. We have now added this discussion to the manuscript (P16).

2. To illuminate the proposed different roles of Rreb1 during mouse gastrulation, a more precise characterization of the cellular phenotypes and molecular defects in the Rreb1 mutants in different region of the epiblast would be required.

We currently show in the manuscript that Rreb1^-/-^ embryos generate the three embryonic germ layers, the ectoderm, mesoderm, and definitive endoderm based on a combination of a panel of markers including *SOX2*, BRACHYURY, GATA6, SOX17. Based on the reviewer’s comments, and our observation that many of the transcriptionally altered genes in mutant embryos are specifically expressed within the endoderm, we now performed additional detailed analysis of definitive endoderm (DE) specification, which occurs at the anterior PS, and endoderm morphogenesis in Rreb1^-/-^ embryos. We found that, while cells expressing the DE marker SOX17 (a marker of nascent DE) were present, they co-expressed BRACHYURY, a marker of the primitive streak and mesoderm. Moreover, these cells exhibited a delay in intercalation into the outer endoderm layer, which likely underpins a number of the morphological defects that we previously described, including the accumulation of cells at the anterior region of the embryo. This new data has now been added in Figure 5F-H and Figure 5 —figure supplement 1C.

Moreover, in the previous (submitted) version of the manuscript, we described a change in the localization of E-CADHERIN protein within the epiblast of Rreb1^-/-^ embryos. We now characterized this phenotype in greater positional detail, showing that this is observed in the proximal but not distal regions of the epiblast. These data have been added to Figure 4 —figure supplement 1F.

What are the identities of the epiblast cells that exhibit ectopic mesenchymal morphology, expression of SNAIL and precious exit from the epithelium? Some of such abnormally behaving cells expressed the epiblast marker SOX2, what makes the authors conclude that they continue to exhibit a more pluripotent epiblast fate and have not acquired mesodermal or endodermal fates. However, this should be investigated with additional mesendodermal markers and Nodal activity reporters. This is important, as the current focus of the manuscript is on the effect Rreb1 LOF has on expression of cytoskeletal and ECM components. Bulk RNA seq experiment are unlikely to reveal subtle changes in expression of SNAIL, or other mesendodermal markers that are expressed in primitive streak. A more thorough characterization of Sox2 Hi cells and "delaminating" cells will also be important, as will clarify apparent differences between Rreb1 mutant and chimeric embryos. Do the observed morphogenetic phenotypes occur without any changes in cell fate specification?

We thank the reviewers for their comments. No good signaling reporters available to assess Nodal activity in vivo in mouse embryos without perturbing the levels of Nodal (for a more detailed discussion see following Essential Revision #3). However, we were able to further investigate the lineage identity of the described ectopic SOX2^HI^ cells by performing immunostaining analysis of a panel of additional markers. We found that SOX2^HI^ cells also continue to express other pluripotency-associated markers including NANOG and OCT4. This led us to ask whether these cells maintained an epiblast-like identity. However, we made the striking observation that, as cells began to exit the organized epiblast epithelium, they downregulated the epiblast marker OTX2. Thus, these cells are not maintained in a pluripotent state. OTX2 is also expressed within the mesoderm and the visceral endoderm (VE) as well as the epiblast at this time. Thus, the absence of OTX2 revealed that ectopic SOX2^HI^ cells do not differentiate towards mesoderm or VE upon epiblast exit.

It was recently shown that, during gastrulation, OTX2 suppresses a primordial germ cell (PGC) identity (J. Zhang et al., 2018). We therefore asked whether, in the absence of OTX2, the ectopic cells that we were observing had adopted a PGC-like identity. PGCs express many pluripotency markers including *SOX2*, NANOG and OCT4, but not the naïve pluripotency marker KLF4. Consistent with this, we found that these ectopic cells did not express KLF4. Furthermore, they had upregulated the PGC marker AP2γ. The marker profile: *Sox2Sox2*^+^ OCT4+ NANOG+ AP2γ+ KLF4- OTX2- is only found within PGCs at this stage of development and therefore suggests that ectopically-positioned cells were exiting pluripotency and acquiring a PGC identity. This information has now been added to the manuscript (Figure 6 —figure supplement 2).

While the ectopic cells observed in chimeric embryos containing Rreb1^-/-^ cells appear to represent a single population, in Rreb1^-/-^ embryos, we observed ectopically positioned cells less frequently and found examples of aberrantly-positioned cells expressing both the pluripotency factor *SOX2*, as well as the mesoderm and endoderm marker GATA6, but not *SOX2*. Thus, in Rreb1^-/-^ embryos, multiple cell types exhibit abnormal invasive-type behaviors. As we observed ectopic Rreb1^-/-^ cells infrequently in the mouse line, perhaps in part as they did not represent a single population, and hence we did not have good markers to identify them by, this precluded further in-depth analysis. However, we have now added further discussion of this within the text (see P13 and P15).

“In Rreb1^-/-^ mutant embryos and chimeras, pluripotent epiblast cells fell out of their epithelial layer into the space between the epiblast and VE. These events were observed more frequently in chimeras versus Rreb1^-/-^ embryos hence, interactions between wild-type and mutant cells, such as differential cell adhesion between these genotypes, may promote invasive-like behaviours. In support of this hypothesis, mathematical models predict that populations with elevated cellular adhesion heterogeneity will exhibit increased tumour cell dissemination (Reher, Klink, Deutsch, and Voss-Bohme, 2017).”

3. No specific molecular link is demonstrated between Rreb1 and any of the reported phenotypes. However, in the Discussion the authors refer to their previous work "We previously showed that, in a cancer model, Rreb1 directly binds to the regulatory region of Snai1 in cooperation with TGF-β activated SMAD transcription factors to induce the expression of SNAIL, which drives EMT (Su et al., 2020)." Yet, in the mouse Rreb1 mutant epiblast, ectopic expression of SNAIL is observed. As Rreb1 can act as transcriptional activator and repressor, such distinct activity in the presence or absence of TGFb-signaling is plausible and experiments testing such models would provide mechanistic insights into these varied functions of Rreb1 during mouse gastrulation. This also underscores the need to more carefully evaluate Nodal signaling and cell fate specification in Rreb1 mutants.

It is unclear what Essential Revision the reviewers are asking in this particular point.

Unfortunately, in the absence of a working anti-RREB1 antibody, it is not possible to do in vivo ChIP on embryos to determine which of the observed transcriptional changes are based on direct interactions with RREB1. Nevertheless, here we demonstrated a link between a change in the expression of a multitude of cytoskeleton-associated factors and the organization of the cytoskeleton as well as associated adherens junctions, which may underpin the loss of epithelial architecture observed.

The reviewer’s asked about potential changes in Nodal signaling activity within Rreb1 mutant embryos. Unfortunately, there are no Nodal signaling activity reporters that do not also perturb Nodal expression. The most commonly used reporter is a Nodal-LacZ allele (Collignon et al., Nature 1996), which as Nodal is itself a target of Nodal signaling, may act as a readout of activity. However, this reporter is a loss of function allele and, as Rreb1 acts downstream of the Nodal pathway, combining these mouse lines would result in genetic interactions that would be difficult to interpret. Other Nodal transcriptional reporters have been designed using specific cis-regulatory elements (Granier et al., Dev. Bio. 2011). However, these do not capture all sites and dynamics of Nodal expression and therefore it is unlikely that these would give a clear answer to the reviewer’s question. As an alternative, we attempted immunofluorescent staining for phosphorylated SMAD2/3 and SMAD2 and SMAD3 using several protocols and antibodies, but were unsuccessful on mouse embryos at this stage of development.

Therefore, in an attempt to address some of the reviewer’s questions regarding Nodal signaling activity in Rreb1^-/-^ embryos, we analyzed the expression levels of a panel of 33 direct targets of Nodal signaling, identified from ESC cultures and mouse embryos (Guzman-Ayala et al., 2009). We found that none of these genes showed a significant change in expression in Rreb1 mutant versus wildtype embryos in our RNA-sequencing dataset (see Author response image 1). Based on this, overall Nodal signaling activity does not appear to be altered in mutant embryos.

While this analysis does not preclude a change in the spatial activity of Nodal signaling in mutant embryos, we believe that the analysis of Nodal reporters on a Rreb1^-/-^ background is beyond the current scope of this paper.In our previous study (Su et al., 2020), we showed that Rreb1 is at the interface of Ras/MAPK and TGF-β signaling. While the reviewers focused specifically on Nodal signaling activity, for which there are no appropriate tools, we have previously developed and characterized a MAPK signaling activity reporter (Spry4^H2BVenus^) (Morgani et al., Dev. Bio. 2018) that we could utilize to determine whether the MAPK signaling response is altered in Rreb1 mutant embryos during gastrulation. We did not observe a significant change in reporter levels at either E6.0 (just prior to the onset of gastrulation) or E7.5 (during gastrulation) (see Author response image 2). However, as we have already extensively characterized multiple aspects of this mutant within the manuscript, we believe that adding this addition data on signaling activities would overcomplicate the story.

**Author response image 2. respfig2:** 

4. The precocious delamination of epiblast cells from epithelia is associated with abnormal ECM organization and broken down basement membrane in ectopic locations in addition to the primitive streak. Kyprianou et al., Nature, 2020 showed that the basement membrane remodeling in the primitive streak region is associated with expression of MMP2 and MMP14 downstream of Nodal signaling. Is expression of these enzymes affected in Rreb1 mutant gastrulae?

Based on this suggestion from the reviewers, we analyzed the expression of Mmp family genes within our RNA-sequencing dataset. We found that *Mmp2*, Mmp11, Mmp14, Mmp15 and *Mmp16* were expressed at robust levels within gastrulating mouse embryos but none of these genes showed a significant difference in their transcription in wildtype versus Rreb1^-/-^ embryos (see Author response image 3).

**Author response image 3. respfig3:** 

We also performed immunofluorescence staining using an anti-*MMP2* antibody but likewise observed no significant difference between wildtype and Rreb1 mutant embryos (see Author response image 4, n = 4 wildtype, n = 5 mutant embryos).

**Author response image 4. respfig4:** 

5. As discussed above, Rreb1 mutant embryos and Rreb1/WT chimeras appear to exhibit distinct phenotypes. Can the authors explain or speculate about these phenotypic differences between the two conditions?

The Rreb1 mutant embryos and chimeras containing Rreb1^-/-^ cells share many phenotypic similarities (see response to Essential Revision #1). However, in Su et al., we highlighted Rreb1^-/-^ chimeric embryos that exhibited an accumulation of cells at the primitive streak, suggesting defects in cell migration away from this region, and consistent with in vitro data suggesting that Rreb1 regulates EMT. In Rreb1^-/-^ embryos, we observed cells accumulating at the primitive streak but with reduced frequency. Differences in the severity and/or penetrance of phenotypes in chimeric embryos versus mutant embryos may arise for various reasons including the fact that in chimeric embryos the embryonic but not extraembryonic supporting tissues are mutant, there is a variable level of contribution of mutant cells in chimeras, and interactions between wildtype and mutant cells that must be considered. This may be particularly pertinent with Rreb1^-/-^ cells, which exhibit an altered localization of cytoskeleton and cell adhesion molecules, and therefore differential adhesion between wildtype and mutant cells is a property of chimeric but not mutant embryos that could drive distinct cell behaviors. We have now added this discussion to the text (P15).

Furthermore, it has been well documented that genetic background affects the expressivity and penetrance of mutant phenotypes. Chimera assays were performed on an inbred genetic background, by injecting 129 embryonic stem cells into C57BL/6 host embryos, whereas our Rreb1^-/-^ analysis in this study was performed largely on an outbred CD1 background, more representative of the genetic diversity within the human population. Inbred genetic backgrounds typically exhibit more severe phenotypes. To briefly assess a potential effect of genetic background on Rreb1^-/-^ phenotypes, we established and analyzed a litter of C57BL/6 Rreb1^-/-^ mice. We found that 4/4 C57BL/6 Rreb1^-/-^ embryos showed defects in exit of cells from the posterior epiblast and breakdown of the LAMININ basement membrane at the primitive streak, consistent with EMT defects observed in chimeras and in vitro in embryoid bodies. Thus, while the phenotypes observed in chimeras versus mutant embryos are comparable, differences in penetrance and expressivity are observed in part due to differences in genetic background as is often the case with studies in the mouse model. We have now added these new data to the paper (see Figure 5 —figure supplement 2).

The authors should determine/report whether breaches of the basement membrane and ECM tracks are observed in Rreb1 mutants similar to chimeric embryos. One would expect to see these near the chains of cells that cross boundaries between germ layers, for example.

As previously discussed, the frequency of ectopic cells in Rreb1^-/-^ embryos is low. Unfortunately, we did not observe these in the batches of embryos that we stained for basement membrane components precluding this further analysis.

5. Interesting vascular phenotypes have been described in the yolk sac, but their cellular and molecular mechanisms are not clear. It would be important to understand the common and unique cellular endpoints that are regulated by this interesting protein. This should be at least discussed.

Based on this suggestion from the reviewers, we investigated the early specification of blood and endothelial progenitors in E7.5 (data not shown) and E8.0 Rreb1^-/-^ embryos, a time when the rudimentary circulatory system is first established, by immunostaining for FLK-1, a factor that is expressed within early hematopoietic and endothelial precursors, and PECAM-1 that is expressed within mature endothelial cells. We observed no obvious difference in the number or localization of FLK-1+ cells within wildtype and Rreb1^-/-^ embryos. This data has been added to Figure 3 —figure supplement 1F and G.

However, as reported in the previous version of the manuscript, by E9.5, we observed a variety of cardiovascular defects. We now added additional characterization of these defects by generating transverse cryosections of these embryos. We found that defects in vascular remodeling were particularly apparent within the dorsal midbrain region. While in wildtype embryos, a thin layer of interconnected FLK1+ PECAM-1+ cells formed within the cephalic mesenchyme, between the neuroepithelium and surface ectoderm layers, in Rreb1 mutant embryos, we instead observed discrete, rounded clusters of these cells. Similar defects were also observed in other regions on the embryo and we additionally noted morphological defects in the dorsal aorta. These new data suggest that the blood and vasculo endothelial phenotypes observed at later developmental stages are not a result of perturbed initial cell fate specification events but are more likely associated with morphogenetic defects. This is consistent with defects resulting from loss of Rreb1 elsewhere in the embryo such as within the epiblast, which is specified correctly but over time structurally degenerates. We have now added these new data to Figure 3F and G and Figure 3 —figure supplement 1K.

Intriguingly, the morphology of Rreb1^-/-^ FLK-1+ PECAM-1+ clusters was similar to that of the extraembryonic mesoderm-derived blood islands within the yolk sac. Formation of the blood islands is driven by paracrine interactions with the VE-derived yolk sac endoderm. Thus, it is tempting to speculate that the delay that we see in dispersal of the VE overlying the epiblast could result in prolonged exposure of the embryo proper to these signals and ectopic blood island formation. Future studies, perhaps employing single-cell RNA sequencing, will generate result in insights into the precise nature of these cells. We have also added further speculation of the underlying causes of these phenotypes and their relation to other mutant phenotypes at multiple places throughout the discussion (P15, and P16).

Furthermore, we identified additional vasculo endothelial-associated factors that were significantly transcriptionally altered within Rreb1 mutant embryos from our RNA-sequencing data. Namely, Bmper, Cxcl10 and H2-Q2. We have also now added this information to the manuscript (P8 and Figure 4 —figure supplement 1A).

6. Based on Figure 1B in which the authors investigate Rreb1 expression in pluripotent ES and epiblast stem cells, it is concluded that "Rreb1 is expressed in the embryonic lineages as pluripotency is exited and the germ layers are specified". Can this be concluded given the strong Rreb1 expression seen in the ICM of pre-implantation stage embryos? More nuanced conclusion should be considered.

We thank the reviewer for drawing attention to this discrepancy in the text. We have now adjusted this and also reference the earlier expression of Rreb1 during pre-implantation stages of mouse embryo development (P5).

“Thus, Rreb1 is expressed within all lineages of the pre-implantation blastocyst but downregulated by the epiblast as it transitions from a naïve to a primed state of pluripotency. During post-implantation development, Rreb1 continues to be expressed in extraembryonic tissues and is re-expressed in the embryonic lineages as primed pluripotency is exited and the germ layers are specified.”

Furthermore, we have altered the title of this Results section to make this clear: “Rreb1 is expressed as cells exit the primed pluripotent state”

7. Rreb1 was shown to promote Snai1 expression in Su et al., yet here Snai1 is ectopically expressed upon loss of Rreb1. The authors speculate that changes in epithelial architecture within the VE of Rreb1 mutants leads to non-specific antibody staining (as with N-cadherin, Figure 5). Is it possible that the very bright Snai1 staining in isolated cells leaving the epiblast (Figure 6) is similarly non-specific dues to localized disruption of epithelial architecture? Is Snai1 staining observed at sites of basement membrane breach in chimeric embryos?

First, it is worth noting that we also observed ectopic expression of Snai1 in chimeric embryos containing Rreb1 mutant cells (see Su et al., Figure 4g, i, Extended Data Figure 10e, f and Essential Revision #1). As such, the data presented here are not inconsistent, and this in agreement with, and corroborating observations from, our previous study.

In addition, when performing immunofluorescence on wildtype gastrulation stage embryos, we (and others) routinely observe high levels of non-specific staining within the outer VE layer at around E6.5 with a variety of antibodies, before the definitive endoderm has intercalated into the VE. This can be identified as strong, non-nuclear staining for a wide variety of transcription factors. We and others have routinely observed this phenomenon, and we have reported this in previous studies (Morgani, Metzger, Nichols, Siggia, and Hadjantonakis, 2018) and believe that it is associated with the highly vacuolated nature of VE cells. As the definitive endoderm intercalates into and disperses the outer VE layer, this non-specific staining is no longer observed in wildtype embryos. In this study non-specific staining observed within the VE can be observed for a number of markers, for example BRACHYURY (Figure. 5D) and N-CADHERIN (Figure 5E), in Rreb1^-/-^ embryos. We hypothesize that this non-specific staining may persist due to the observed developmental delay in Rreb1^-/-^ affecting the timing of definitive endoderm intercalation (new data added to Figure 5F-H and Figure 5—figure supplement 1C and discussed in Essential Revision #2).

In contrast to the non-nuclear staining observed in the outer VE layer, SNAIL staining within the embryo-proper was almost entirely nuclear-localized as expected for a transcription factor. Furthermore, we did not see ectopic staining within the epiblast of a large number of other lineage-specific markers that we analyzed within mutant embryos. Thus, there is no reason to believe that the SNAIL staining we observed is non-specific.

8. In the discussion (page 14, lines 25-27), the authors speculate that Rreb1 expression within the VE may be important for epithelial integrity in the epiblast… but isn't this inconsistent with results from chimeric embryos, in which WT VE does not prevent ectopic delamination of mutant (and WT) epiblast cells?

We thank the reviewers for drawing out attention to this. We have now removed this from the text and instead added discussion about how indirect changes in the mechanical forces experienced by the epiblast could lead to these phenotypes (P17):

“As Rreb1 is not expressed highly within the epiblast, these phenotypes could be due to a loss of low-level epiblast expression or an indirect effect of altered mechanical forces in the embryo stemming from perturbed EMT and, in mutant embryos, the VE. The expression of SNAIL and a number of other EMT and adhesion regulators is mechano-sensitive (Farge, 2003; Pukhlyakova, Aman, Elsayad, and Technau, 2018; K. Zhang et al., 2019), and thus changes in the physical forces within the embryo could underpin ectopic SNAIL expression within a fraction of epiblast cells.”

9. For several phenotypes, including notochord defects (Figure S2), punctate localization of F-actin and E-cadherin (Figure 4), and Afp-GFP localization (Figure 3), only 1 example is shown and no sample numbers are reported, making it difficult to evaluate how representative these images are of mutant phenotypes. The authors should include the number of embryos examined for every experiment shown.

Embryo numbers are presented on the graphs for which quantification was performed e.g. scoring of embryonic lethality at different stages, embryo length, somite number etc. We have now added “n” = embryo numbers for immunofluorescence analyses into the figure legends and/or in the body of the text. In light of the reviewer’s comments, we have also included additional embryo images, where space permits, including examples of punctate E-CADHERIN localization within different regions of the embryo (Figure 4 —figure supplement 1F) and Afp-GFP expression in mutant embryos (Figure 3 —figure supplement 1D).